# STRUCTCHART: PERCEPTION, STRUCTURING, REASONING FOR VISUAL CHART UNDERSTANDING

## ABSTRACT

Charts are common in literature across different scientific fields, conveying rich information easily accessible to readers. Current chart-related tasks focus on either chart perception which refers to extracting information from the visual charts, or performing reasoning given the extracted data, *e.g.* in a tabular form. In this paper, we aim to establish a unified and label-efficient learning paradigm for joint perception and reasoning tasks, which can be generally applicable to different downstream tasks, beyond the question-answering task as specifically studied in peer works. Specifically, StructChart first reformulates the chart information from the popular tubular form (specifically linearized CSV) to the proposed Structured Triplet Representations (STR), which is more friendly for reducing the task gap between chart perception and reasoning due to the employed structured information extraction for charts. We then propose a Structuring Chart-oriented Representation Metric (SCRM) to quantitatively evaluate the performance for the chart perception task. To enrich the dataset for training, we further explore the possibility of leveraging the Large Language Model (LLM), enhancing the chart diversity in terms of both chart visual style and its statistical information. Extensive experiments are conducted on various chart-related tasks, demonstrating the effectiveness and promising potential for a unified chart perception-reasoning paradigm to push the frontier of chart understanding.

## 1 INTRODUCTION

Charts are common tools for visualizing large amounts of information. Automatically extracting the underlying information from the visual charts has become an emerging research topic in learning (Masry et al., 2022; Nam et al., 2023) and vision communities (Luo et al., 2021; Obeid & Hoque, 2020; Rane et al., 2021), which can ultimately help better acquiring the data from the existing massive multi-modal corpus beyond raw texts or numbers.

Visual Chart Understanding (CU) aims to extract the statistical information contained within a given visual chart and perform the corresponding downstream tasks (*e.g.*, chart question answering or chart redrawing) according to the extracted information, which is practical in many fields including medical tabular analysis (Ulmer et al., 2020), chart Optical Character Recognition (OCR) (Luo et al., 2021; Hegselmann et al., 2023; Obeid & Hoque, 2020; Masry et al., 2022), and knowledge data extraction for Large Language Models (LLMs) (Brown et al., 2020; Chung et al., 2022; He et al., 2022). Recently, chart-related research works can be categorized into two classes (1) Chart Perception (CP) class that focuses on recognizing valuable information from a chart, converting the chart from a visual-level image to text-level representation, and (2) Chart Reasoning (CR) class that aims to understand the chart information by a tabular form. Although these works (Masry et al., 2022; Raffel et al., 2020; Luo et al., 2021; Chung et al., 2022) are inspiring and have achieved promising performance gains on chart perception or chart reasoning task, joint perception-reasoning is still under-explored and challenged by the following aspects.

**(1) Large perception-reasoning task gap:** Perception task tries to extract as accurate chart information as possible, ignoring the subtle relations between data columns and rows. However, the task of reasoning is often required to consider the complicated data relations to output the right answer or summarize the chart information, especially for chart data that combine both numerical and textual information. **(2) Incomplete metric evaluation:** There lacks a comprehensive metric to evaluate chart perception performance from the perspective of structured information extraction with data

relations. Besides, the current performance metric (Masry et al., 2022) only covers a single type of chart data such as bars (Choi et al., 2019), pie (Liu et al., 2019), and line (Luo et al., 2021), which is hard to be generalized to different chart domains when scaling up the number of chart data. **(3) Expensive chart data:** Acquiring charts from different fields and manually annotating these charts are highly dependent on professionals from different fields, which makes the chart data acquisition and annotation more difficult, labor-intensive, and time-consuming (Ulmer et al., 2020).

To tackle the above challenges, we propose StructChart, a novel approach for a unified and label-efficient learning paradigm for joint perception and reasoning tasks. Firstly, to **alleviate the task gap**, StructChart adopts an image-encoder and text-decoder to facilitate the representation transformation from chart images to text format using Linearized Comma-Separated Values Tokens (LCT). But we argue that LCT ignores the entity relation within the chart, and thus, propose to reformulate the chart from the commonly-used LCT format to a well-designed Structured Triplet Representations (STR) format. Secondly, to **unify the metric evaluation**, we develop a Structuring Chart-oriented representation Metric (SCRM) based on the proposed STR, which evaluates the chart perception ability from the STR (structured information description), enabling the perception evaluation process for different types of chart data. Finally, to **expand the chart data**, we propose an LLM-based self-inspection data production scheme that generates more chart data with different domain distributions by statistical data query and drawing code generation leveraging the LLMs. We found that the chart perception and reasoning ability can be enhanced by the LLMs-based simulation method.

Experiments are conducted on both chart perception and reasoning tasks, including chart perception, chart question answering, chart summarization, and chart redrawing. Besides, we produce a synthetic chart dataset termed SimChart9K. We observe that the SimChart9K significantly boosts the chart perception performance, even obtaining a high performance under the few-shot condition. Overall, experimental results verify that the proposed StructChart paradigm is able to achieve a high-performance chart perception and unify the chart understanding.

**Contribution.** **(1)** For robust chart perception and reasoning, we propose the so-called Structured Triplet Representations (STR), which replaces the widely-used linearized CSV tokens for chart-related tasks. **(2)** Based on the proposed STR format, we design a novel Structuring Chart-oriented Representation Metric (SCRM) applicable to various chart-related perception tasks whose evaluation sensitivity can be flexibly tuned by a preset hyper-parameter. **(3)** We perform data augmentation for chart perception and reasoning by leveraging an LLMs-based self-inspection data production scheme, producing the SimChart9K dataset. Besides, we observe that StructChart continuously improves the chart perception performance as more simulated charts are used for pre-training.

## 2 RELATED WORKS

Our work is focused on Chart Understanding (CU). We discuss works in this emerging area in several aspects, and leave the literature on Vision Language Pre-trained Models (VLPMs) in Appendix A.

**Chart Perception** refers to obtaining the numerical and textual values (often in the tabular) from the charts. ChartReader (Rane et al., 2021) takes a combined approach, using rule or heuristic-based edge extraction supported by OCR for text elements. (Choi et al., 2019) adopted the idea of general object detection to detect the bar components by treating each bar as an object. ChartOCR (Luo et al., 2021) employs a modified version of CornerNet (Law & Deng, 2018) backbone for keypoint detection to reconstruct the chart components (*e.g.* bars and sectors), and OCR for component value.

**Chart Reasoning** seeks to leverage chart image information in order to execute logical or mathematical reasoning processes, where Question Answering (QA) is a representative task for showing the chart reasoning ability. VL-T5-OCR (Masry et al., 2022) and VisionTaPas-OCR (Masry et al., 2022) extend cross-modality encoder in T5 (Raffel et al., 2020) and TaPas (Herzig et al., 2020) to consider chart image features. Besides, Pix2Struct (Lee et al., 2022) tries to use the screenshot parsing input to perform the self-supervised pre-training from abundant website data.

**Chart Understanding** is at a wider level than chart reasoning (at least by the scope of this paper), covering more open-ended and high-level tasks. Besides question answering task, chart understanding contains a wider variety of generative tasks, such as chart summarization, chart redrawing, *etc*. Matcha (Liu et al., 2022b) and Deplot (Liu et al., 2022a) are the pioneering attempts for chart understanding, with both carrying out the QA and summarization tasks. Matcha (Liu et al., 2022b) pre-trains a Pix2Struct (Lee et al., 2022) with chart derendering and math reasoning tasks, while

Table 1: Comparisons of different research works on chart data, where CP, CR, and CU represent the chart perception, chart reasoning, and chart understanding works, respectively. S. and R. denote the summarization and redrawing downstream task, respectively.

| | Methods | Chart Types | | | Perception | Reasoning / Understanding | Perception | | Downstream Tasks | | |
|---|---|---|---|---|---|---|---|---|---|---|---|
| | | Line | Bar | Pie | | | Format | Metric | QA | S. | R. |
| CP | ReVision (Savva et al., 2011) | | ✓ | ✓ | ✓ | | | | | | |
| | ChartReader (Rane et al., 2021) | | ✓ | | ✓ | \ | JSON | Component-level | \ | | |
| | (Liu et al., 2019) | | ✓ | | ✓ | | | | | | |
| | (Choi et al., 2019) | | | ✓ | ✓ | | | | | | |
| | ChartOCR (Luo et al., 2021) | ✓ | ✓ | ✓ | ✓ | | | Type-level | | | |
| CR | T5-OCR (Masry et al., 2022) | ✓ | ✓ | ✓ | ✓ | ✓ | LCT | \ | ✓ | | |
| | TaPas-OCR (Masry et al., 2022) | ✓ | ✓ | ✓ | ✓ | ✓ | | | ✓ | | |
| | VL-T5-OCR (Masry et al., 2022) | ✓ | ✓ | ✓ | ✓ | ✓ | | | ✓ | | |
| | VisionTaPas-OCR (Masry et al., 2022) | ✓ | ✓ | ✓ | ✓ | ✓ | | | ✓ | | |
| CU | Matcha (Liu et al., 2022b) | ✓ | ✓ | ✓ | − | ✓ | − | − | ✓ | ✓ | |
| | Deplot (Liu et al., 2022a) | ✓ | ✓ | ✓ | ✓ | ✓ | LCT | − | ✓ | ✓ | |
| | StructChart (Ours) | ✓ | ✓ | ✓ | ✓ | ✓ | STR | SCRM | ✓ | ✓ | ✓ |

Deplot (Liu et al., 2022a) harnesses Vision Language Pre-trained Model (VLPM) to extract chart information, and subsequently employs LLMs to conduct inference for the QA and summarization.

Overall, Table 1 summarizes the differences between our StructChart and other chart-related works.

## 3 THE PROPOSED METHOD

StructChart includes four key components: ***(1) Transformer-based Chart-oriented Information Extractor (CIE)***. It incorporates an image-encoder and text-decoder to facilitate the transformation from chart images to text format using Comma-Separated Values (CSV). ***(2) Structured Representation Transformation.*** The extracted intermediate CSV text is structured into a triplet form to elucidate the intricate position relationship between the header and index. ***(3) Structuring Chart-oriented Representation Metric (SCRM).*** We further design a metric that comprehensively evaluates the quality of the transformed triplets, which facilitates the subsequent reasoning. ***(4) LLM-based Self-inspection Data Production Scheme.*** We devise a novel methodology for chart data simulation to enhance zero-shot and few-shot perception and reasoning ability, achieving continuous performance gains when scaling up the simulated charts. The whole paradigm is illustrated in Fig. 1.

### 3.1 DESIGN FOR TWO-STAGE STRUCTCHART

Different from end-to-end multi-modal reasoning tasks (Masry et al., 2022; Obeid & Hoque, 2020), we fulfill CU by solving two independent tasks: perception and reasoning, with the Structured Triplet Representations (STR) data representation serving as a bridge between them.

**Perception Stage.** We propose a CIE that utilizes a pixel-level encoder and text-level decoder, both based on the Vision Transformer (ViT) (Dosovitskiy et al., 2020). Instead of standard ViT, which scales the input images to a predefined resolution, we propose to always scale the input image to a fixed number of patches that can fit within the longest given sequence length, according to the original resolution of the input image. Moreover, we add 2-dimensional absolute positional embeddings for the input patches, allowing the perception module to handle variable resolutions. At this stage, the chart at pixel level can be converted to text-level Linearized CSV Tokens (LCT).

**Reasoning Stage.** Before performing the reasoning process, we structure the LCT into the designed STR to facilitate the module's understanding of chart-oriented information (see Sec. 3.2 for details). This structuring process enables us to reason over the text, providing a better understanding of the entity relation within a chart. Considering the difficulty of evaluating downstream tasks, the reasoning process is performed on the QA task using GPT-3.5 (Brown et al., 2020) without any prompt engineering strategies. Additionally, we verify the effectiveness of the proposed STR and compare with previous research works (Luo et al., 2021; Masry et al., 2022; Liu et al., 2022a;b).

The motivations of the designed two-stage pipeline are: 1) explicit chart image representations can enhance the interpretability of the subsequent reasoning stage, and 2) the extracted STR/LCT can be used as the pre-training corpus for large language models and vision-language models.

### 3.2 STRUCTURED TRIPLET REPRESENTATIONS (STR) FOR CHART UNDERSTANDING

Visual charts often contain rich textual and numerical information. Generally, the chart information is represented by long-form texts in the form of CSV, *i.e.* previously-mentioned LCT. However, the LCT format is sensitive to positional variations of entities from charts due to that it is used in a linear form. Thus, we propose to reformulate the LCT format in order to effectively and robustly represent the positional relations between row and column headers of a given chart.

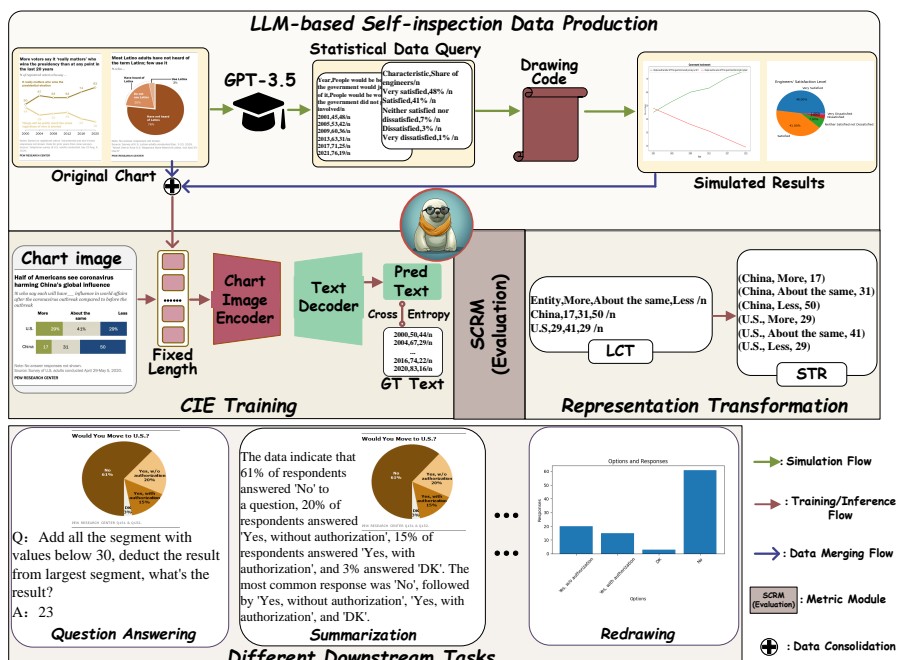

Figure 1: StructChart overview: (1) LLM-based production scheme for providing more chart data; (2) CIE training for chart perception; (3) Representation transformation for bridging the task gap; (4) Downstream reasoning tasks, including Question Answering, Summarization and Redrawing.

**Task Definition and Structuring.** Given a chart image, the extracted LCT can be described as:

$$
\begin{aligned}
C_{csv} := none, \quad & Entity_{c_1}, Entity_{c_2}, ..., Entity_{c_m}, ...Entity_{c_M} \ /n \\
& Entity_{r_1}, Value_{r_1}^{c_1}, Value_{r_1}^{c_2}, ..., Value_{r_1}^{c_m}, ..., Value_{r_1}^{c_M} \ /n \\
& ...... \\
& Entity_{r_n}, Value_{r_n}^{c_1}, Value_{r_n}^{c_2}, ..., Value_{r_n}^{c_m}, ..., Value_{r_n}^{c_M} \ /n \\
& ...... \\
& Entity_{r_N}, Value_{r_N}^{c_1}, Value_{r_N}^{c_2}, ..., Value_{r_N}^{c_m}, ..., Value_{r_N}^{c_M} \ /n \ ,
\end{aligned}
\tag{1}
$$

where $/n$ refers to line break, and $Entity_{r_n}$ and $Entity_{c_m}$ indicate $n$-th row header entity and $m$-th column header entity, respectively, where $(M, N \in \mathbb{N}^+)$. $Value_{r_n}^{c_m}$ in Eq. 1 contains the positional information of $Entity_{c_m}$ and $Entity_{r_n}$. However, the LCT still faces two issues: *(1) The evaluation process for the predicted long-form texts containing positional information for perception model selection is non-trivial. (2) Highly position-sensitive LCT format increases the inference difficulty of different downstream chart tasks.*

Considering that chart information has matrix-like row-column transformation invariance and transpose transformation invariance, the LCT is structured into a well-designed triplet, with higher granularity for evaluation and downstream. Given LCT tokens $C_{csv}$ shown in Eq. 1, the structured triplet representations can be obtained as follows:

$$
\begin{aligned}
C_{tri} := & (Entity_{r_1}, Entity_{c_1}, Value_{r_1}^{c_1}), \\
& (Entity_{r_1}, Entity_{c_2}, Value_{r_1}^{c_2}), \\
& ........, \\
& (Entity_{r_n}, Entity_{c_m}, Value_{r_n}^{c_m}), \\
& ........, \\
& (Entity_{r_N}, Entity_{c_M}, Value_{r_N}^{c_M}).
\end{aligned}
\tag{2}
$$

**Evaluation Metric Design.** Furthermore, we design a **Structuring Chart-oriented Representation Metric (SCRM)** to comprehensively evaluate the extracted chart information represented using the proposed STR. When comparing the predicted STR and Ground Truth (GT) STR $C_{tri}$, we treat $Entity_{r_n}, Entity_{c_m}$ as strings and $Value_{r_n}^{c_m}$ as floats, respectively. More detailed description of the matching process is provided in Appendix B.

**1) Image-wise**. Suppose that there are totally $\mathbf{P}$ triplets from the model prediction and $\mathbf{Q}$ triplets from GT ($\mathbf{P}, \mathbf{Q} \in \mathbb{N}^+$), the evaluation process is shown as follows:

- For $Entity$, we obtain the edit distance of the $p$-th prediction string and the $q$-th GT string:

$$J(p,q) = \frac{\mid Entity_{pred}^p \cup Entity_{GT}^q \mid - \mid Entity_{pred}^p \cap Entity_{GT}^q \mid}{\mid Entity_{pred}^p \cup Entity_{GT}^q \mid} . \tag{3}$$

- For $Value$, we calculate the relative error between the $p$-th prediction value and the $q$-th GT value:

$$e(p,q) = \left| \frac{Value_{pred}^p - Value_{GT}^q}{Value_{GT}^q} \right| . \tag{4}$$

- To achieve a comprehensive evaluation, we design three levels of tolerance for fine-grained judgment, aiming to measure the similarity between the predicted triplets and GT triplets, by calculating the Intersection over Union $IoU|_{tol}$, under the given tolerance level $tol$ as follows:

$$l(p,q)|_{tol} = \begin{cases} 1, & if : J(p,q) \leq J_{thr}|_{tol} \ \wedge \ e(p,q) \leq e_{thr}|_{tol} \\ 0, & else \end{cases}, \tag{5}$$

$$tol := \{strict, slight, high\}, \quad strict := \{J_{thr}|_{tol} = 0 \wedge e_{thr}|_{tol} = 0\},$$
$$slight := \{J_{thr}|_{tol} = 2 \wedge e_{thr}|_{tol} = 0.05\}, \quad high := \{J_{thr}|_{tol} = 5 \wedge e_{thr}|_{tol} = 0.1\}, \tag{6}$$

$$IoU|_{tol} = \frac{\sum_{q=1}^{\mathbf{Q}} \sum_{p=1}^{\mathbf{P}} l(p,q)|_{tol}}{\mathbf{P} + \mathbf{Q} - \sum_{q=1}^{\mathbf{Q}} \sum_{p=1}^{\mathbf{P}} l(p,q)|_{tol}}. \tag{7}$$

**2) Dataset-wise**. Given the dataset with $\mathbf{L}$ chart images ($\mathbf{L} \in \mathbb{N}^+$), the Intersection over Union of the $i$-th image can be denoted as $IoU(i)$. Besides, given a preset similarity threshold $IoU_{thr}$, the corresponding discriminant function towards the positive and negative images can be written as:

$$d(i)|_{IoU_{thr},tol} = \begin{cases} 1, & if : IoU(i)|_{tol} \geq IoU_{thr} \\ 0, & else \end{cases}. \tag{8}$$

When the preset similarity threshold $IoU_{thr}$ becomes a variable (denoted as $t$), it changes to:

$$d(i,t)|_{tol} = \begin{cases} 1, & if : IoU(i)|_{tol} \geq t \\ 0, & else \end{cases}. \tag{9}$$

The proposed metric SCRM consists of two indicators ($Precision$ with a fixed similarity threshold and $mPrecision$ with a varying one in the range $(0.5 : 0.05 : 0.95)$):

$$Precision|_{IoU_{thr},tol} = \frac{\sum_{i=1}^{L} d(i)|_{IoU_{thr},tol}}{L}, \quad mPrecision|_{tol} = \frac{\sum_{t=10}^{19} \sum_{i=1}^{L} d(i, 0.05t)|_{tol}}{10L}. \tag{10}$$

### 3.3 SIMULATING CHARTS WITH ENHANCED DIVERSITY FOR PRETRAINING-FINETUNING

Considering the difficulty and cost of chart data acquisition and labeling, we introduce an LLM-based text-to-chart level data production scheme, dividing into: *(1) statistical data query to ensure the data-level diversity*, and *(2) drawing code generation to ensure the drawing style diversity*. The complete schematic simulation paradigm is shown in Appendix E.

**Statistical Data Query.** Given the chart dataset $\mathbb{D}_{ori} = \{\mathbf{I}_{ori}, \mathbf{T}_{ori}\}$, where $\mathbf{I}_{ori} = \{I_{ori}^1, I_{ori}^2, I_{ori}^3, ..., I_{ori}^n, ..., I_{ori}^N\}$ are the images and $\mathbf{T}_{ori} = \{T_{ori}^1, T_{ori}^2, T_{ori}^3, ..., T_{ori}^n, ..., T_{ori}^N\}$ are the corresponding labeled texts in CSV format. At this stage, we define the task as the imitation of labeled CSV texts $\mathbf{T}_{ori}$ in origin chart dataset $\mathbb{D}_{ori}$ through an LLM (specifically GPT-3.5) to generate simulated CSV labels $\mathbf{T}_{sim}$. In detail, We employ the few-shot prompting, leveraging generative model (Brown et al., 2020) to complete the imitation task. For diversity and effectiveness of the imitation data, we impose three restrictions on the demonstration and instruction: *(1) The simulated content $T_{sim}^n$ must be in CSV format, (2) The scale of $T_{sim}^n$ can be altered compared with $T_{ori}^n$, including the number of rows, and (3) The combination of text in $T_{sim}^n$ must be reasonable, even though it may be highly irrelevant to $T_{ori}^n$.*

**Drawing Code Generation.** Having obtained the simulated texts (in CSV format) $\mathbf{T}_{sim} = \{T_{sim}^1, T_{sim}^2, T_{sim}^3, ..., T_{sim}^n, ..., T_{sim}^N\}$, the next step is to create images $\mathbf{I}_{sim} =$

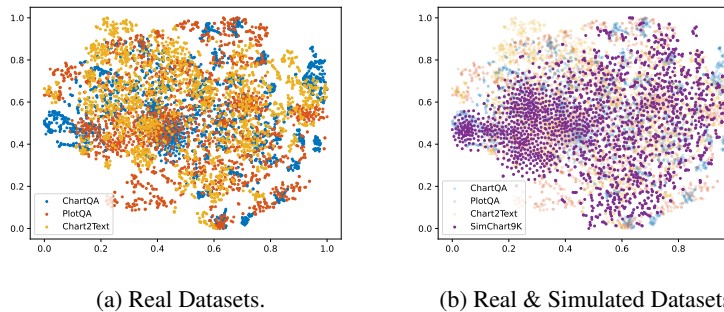

(a) Real Datasets.  (b) Real & Simulated Datasets.

Figure 2: Feature distributions of ChartQA, PlotQA, Chart2Text, and SimChart9K visualized by t-Distributed Stochastic Neighbor Embedding (t-SNE) (Van der Maaten & Hinton, 2008).

$\{I_{sim}^1, I_{sim}^2, I_{sim}^3, ..., I_{sim}^n, ..., I_{sim}^N\}$ based on $\mathbf{T}_{sim}$. We still employ the advanced GPT-3.5 to directly generate the drawing code, and draw the simulated chart $I_{sim}^n$ based on the aforementioned statistical data $T_{sim}^n$, by means of our developed instruction prompting. To guarantee the diversity of chart distribution at the image level, we implement the following limitations within the instruction prompting: *(1) Random selection of a chart type that is appropriate for $T_{sim}^n$*, *comprising histograms, scatterplots, line charts, and pie charts*, *(2) Random choice of drawing style*, *including fonts, colors, line styles, backgrounds, etc*, and *(3) Transformation of scale with different coordinate axes.*

To guarantee that the generated drawing code can be executed correctly, we design a self-inspection mechanism to iteratively skip the non-executable code generated by GPT-3.5, until all the drawing code that matches the corresponding text can be executable. Benefiting from the diversities of the texts generated by GPT-3.5, we can generate more diverse Statistical Data Queries and Drawing Codes, according to the given $T_{ori}^n$. As a result, the proposed LLM-based text-to-chart level data production scheme can be used to simulate scalable, data-rich and style-diverse chart dataset $\mathbb{D}_{sim} = \{\mathbf{I}_{sim}, \mathbf{T}_{sim}\}$ based on scale-invariant or few-shot original chart datasets $\mathbb{D}_{ori} = \{\mathbf{I}_{ori}, \mathbf{T}_{ori}\}$

## 4 EXPERIMENTS

### 4.1 EVALUATION DATASETS AND IMPLEMENTATION DETAILS

**Datasets.** We evaluate our StructChart on three real-world chart benchmarks and our simulated dataset with image-CSV pairs. **ChartQA** (Masry et al., 2022) is a large-scale visual reasoning dataset with 20,882 charts collected from online sources, which can be divided into an augmented set generated synthetically and a human set written by humans. **PlotQA** (Methani et al., 2019) is a synthetic dataset that covers 28.9 million question-answer pairs spread across 224,377 plots. FigureQA (Kahou et al., 2017) is a visual reasoning corpus consisting of over one million human-designed question-answer pairs. **Chart2Text** (Obeid & Hoque, 2020) is a dataset for automatic summarization of statistical charts crawled from statista.com, yielding total 8,305 charts with annotations. **SimChart9K** is the proposed simulated dataset composed of 9,098 charts and associated data annotations in CSV format. We have previously elucidated the simulation methodology in Sec. 3.3, where it was generated via ChartQA using a generative model.

**Implementation and Evaluation Metrics.** We design various data consolidation settings based on both real and simulated data, when training StructChart using the architecture mentioned in Sec. 3.1. For perception, we employ SCRM (Sec. 3.2) for perception model selection, and obtain STR of input chart from the well-trained StructChart model. For reasoning, we use GPT-3.5 without prompt engineering strategy to perform different downstream tasks for fair comparison with other methods.

### 4.2 CHART PERCEPTION RESULTS ON REAL-WORLD AND SIMULATION DATA

We first conduct the experiments on real-world datasets including ChartQA, PlotQA, and Chart2Text, and further merge them for joint-dataset training, to better evaluate the performance scalability given more training data. From Table 2, with a generalizability study conducted on ChartQA, PlotQA and Chart2Text, StructChart continuously improves the perception performance of chart data on each domain given more real or simulated training samples. Moreover, Fig. 2a visualizes the feature distributions via t-SNE (Van der Maaten & Hinton, 2008), where features from different datasets are basically consistent.

Table 2: Results on the validation set of ChartQA, PlotQA, and Chart2Text under: 1) single-set that the model is trained and evaluated on the same dataset, 2) trained on the merging-set that training samples are merged from the real datasets, and 3) trained on the merging-set that training samples are merged from both real and simulated datasets. **Note that** 'Real Merging' is all real data (ChartQA+PlotQA+Chart2Text), 'Real&Sim Merging' refers to both real and simulated data (ChartQA+PlotQA+Chart2Text+SimChart9K), and 'C.D.' denotes Closed-source Dataset.

| Val Set | Model | Train Set | $IoU_{thr} \rightarrow$ Tolerance ↓ | mPrecision 0.5:0.05:0.95 | Precision 0.5 | 0.75 | 0.95 | 1 (EM) |
|---|---|---|---|---|---|---|---|---|
| ChartQA | Matcha (Liu et al., 2022b) | ChartQA+ PlotQA+ C.D. | strict 🟥 | 0.5160 | 0.5814 | 0.5114 | 0.4678 | 0.4460 |
| | | | slight 🟨 | 0.6598 | 0.7045 | 0.6572 | 0.6250 | - |
| | | | high 🟩 | 0.7161 | 0.7519 | 0.7150 | 0.6894 | - |
| | Deplot (Liu et al., 2022a) | ChartQA+ PlotQA+ C.D. | strict 🟥 | 0.6331 | 0.7008 | 0.6326 | 0.5814 | 0.5663 |
| | | | slight 🟨 | 0.7666 | 0.8229 | 0.7661 | 0.7282 | - |
| | | | high 🟩 | 0.8150 | 0.8759 | 0.8087 | 0.7812 | - |
| | Our StructChart | ChartQA | strict 🟥 | 0.6770 | 0.7273 | 0.6714 | 0.6458 | 0.6326 |
| | | | slight 🟨 | 0.7792 | 0.8220 | 0.7746 | 0.7519 | - |
| | | | high 🟩 | 0.8274 | 0.8703 | 0.8210 | 0.8011 | - |
| | Our StructChart | Real Merging | strict 🟥 | 0.7017 | 0.7547 | 0.6998 | 0.6610 | 0.6506 |
| | | | slight 🟨 | 0.8227 | 0.8674 | 0.8201 | 0.7926 | - |
| | | | high 🟩 | **0.8591** | 0.8987 | **0.8551** | **0.8362** | - |
| | Our StructChart | Real&Sim Merging | strict 🟥 | 0.7187 | 0.7683 | 0.7153 | 0.6705 | **0.6642** |
| | | | slight 🟨 | 0.8311 | 0.8761 | 0.8301 | 0.8001 | - |
| | | | high 🟩 | 0.8568 | **0.8990** | 0.8542 | 0.8358 | - |
| PlotQA | Matcha (Liu et al., 2022b) | ChartQA+ PlotQA+ C.D. | strict 🟥 | 0.0048 | 0.0089 | 0.0048 | 0.0036 | 0.0036 |
| | | | slight 🟨 | 0.0752 | 0.0909 | 0.0754 | 0.0635 | - |
| | | | high 🟩 | 0.0823 | 0.1093 | 0.0837 | 0.0719 | - |
| | Deplot (Liu et al., 2022a) | ChartQA+ PlotQA+ C.D. | strict 🟥 | 0.0997 | 0.1532 | 0.1021 | 0.0641 | 0.0629 |
| | | | slight 🟨 | 0.6969 | 0.8664 | 0.7435 | 0.5463 | - |
| | | | high 🟩 | 0.7471 | 0.9679 | 0.8034 | 0.5992 | - |
| | Our StructChart | PlotQA | strict 🟥 | 0.1995 | 0.2500 | 0.1931 | 0.1765 | 0.1736 |
| | | | slight 🟨 | 0.7848 | 0.8519 | 0.7784 | 0.7405 | - |
| | | | high 🟩 | 0.8271 | 0.8922 | 0.8223 | 0.7861 | - |
| | Our StructChart | Real Merging | strict 🟥 | 0.4549 | 0.5855 | 0.4525 | 0.3521 | 0.3385 |
| | | | slight 🟨 | 0.8589 | 0.9210 | 0.8569 | 0.8118 | - |
| | | | high 🟩 | 0.8921 | 0.9466 | 0.8860 | 0.8557 | - |
| | Our StructChart | Real&Sim Merging | strict 🟥 | 0.4596 | 0.5901 | 0.4569 | 0.3563 | **0.3612** |
| | | | slight 🟨 | 0.8612 | 0.9234 | 0.8590 | 0.8138 | - |
| | | | high 🟩 | **0.8998** | **0.9547** | **0.8935** | **0.8591** | - |
| Chart2Text | Our StructChart | Chart2Text | strict 🟥 | 0.1936 | 0.2473 | 0.1892 | 0.1533 | 0.1442 |
| | | | slight 🟨 | 0.5524 | 0.6603 | 0.5529 | 0.4672 | - |
| | | | high 🟩 | 0.6945 | 0.7676 | 0.6934 | 0.6356 | - |
| | Our StructChart | Real Merging | strict 🟥 | 0.3156 | 0.4002 | 0.3123 | 0.2509 | 0.2318 |
| | | | slight 🟨 | 0.7141 | 0.7938 | 0.7205 | 0.6426 | - |
| | | | high 🟩 | 0.8085 | 0.8595 | 0.8090 | 0.7673 | - |
| | Our StructChart | Real&Sim Merging | strict 🟥 | 0.3394 | 0.4261 | 0.3367 | 0.2749 | **0.2635** |
| | | | slight 🟨 | 0.7759 | 0.8522 | 0.7701 | 0.7096 | - |
| | | | high 🟩 | **0.8296** | **0.8791** | **0.8287** | **0.7700** | - |

We randomly divide the proposed SimChart9K dataset into four subsets with different amounts (0.1K, 1K, 6K, 9K), and train the StructChart on the mixed dataset (including ChartQA training set and SimChart with different amounts). All evaluations are conducted on the ChartQA validation set based on the proposed SCRM metrics. In Table 3, it can be seen that the introduction of simulation dataset significantly improves the CIE performance of StructChart; that is, **the larger the simulation dataset, the greater the performance gains in CIE**. Furthermore, we also illustrate feature distributions of SimChart9K in Fig. 2b. It can be observed from Fig. 2b that the feature distribution of the simulated data (SimChart9K) can be better matched with that of the real data such as ChartQA, PlotQA, and Chart2Text. As a result, the simulated charts are beneficial to boost the CIE performance of the model towards the real-world charts.

## 4.3 ACHIEVING 100% PERFORMANCE BY ONLY 20% REAL DATA

The purpose of this part is to answer two questions: 1) Can we achieve a high-performance CIE only leveraging few-shot real samples? 2) With the help of SimChart9K, how many real-world samples can obtain the CIE performance that is achieved on the full training set? To answer these questions, we split the real chart dataset ChartQA into subsets with different sizes, including subsets with 1%, 10%, 20% and 50% original real-world samples. We demonstrate zero-shot and few-shot results in Table 5, obtaining the following observations: (1) When the model is trained on the real dataset ChartQA alone without any simulated samples, the CIE performance is still positively cor-

Table 3: ChartQA perception results by scaling up the simulation data (from 0.1K to 9K).

| Train Set | | $IoU_{thr} \rightarrow$ | mPrecision | Precision | | | |
|---|---|---|---|---|---|---|---|
| | | Tolerance ↓ | 0.5:0.05:0.95 | 0.5 | 0.75 | 0.95 | 1 (EM) |
| Real & Simulated Source | ChartQA (w/o simulation data) | strict ■ | 0.6770 | 0.7273 | 0.6714 | 0.6458 | 0.6326 |
| | | slight ■ | 0.7792 | 0.8220 | 0.7746 | 0.7519 | - |
| | | high ■ | 0.8274 | 0.8703 | 0.8210 | 0.8011 | - |
| | ChartQA+SimChart 0.1K | strict ■ | 0.6804 | 0.7282 | 0.6752 | 0.6468 | 0.6383 |
| | | slight ■ | 0.7893 | 0.8305 | 0.7850 | 0.7595 | - |
| | | high ■ | 0.8326 | 0.8731 | 0.8277 | 0.8087 | - |
| | ChartQA+SimChart 1K | strict ■ | 0.6871 | 0.7367 | 0.6828 | 0.6544 | 0.6458 |
| | | slight ■ | 0.7938 | 0.8371 | 0.7907 | 0.7661 | - |
| | | high ■ | 0.8394 | 0.8788 | 0.8362 | 0.8116 | - |
| | ChartQA+SimChart 6K | strict ■ | 0.7040 | 0.7491 | 0.7036 | 0.6686 | 0.6591 |
| | | slight ■ | 0.8128 | 0.8580 | 0.8116 | 0.7794 | - |
| | | high ■ | 0.8450 | 0.8835 | 0.8428 | 0.8182 | - |
| | ChartQA+SimChart 9K | strict ■ | 0.7116 | 0.7595 | 0.7074 | 0.6809 | **0.6686** |
| | | slight ■ | 0.8182 | 0.8674 | 0.8144 | 0.7850 | - |
| | | high ■ | **0.8527** | **0.8958** | **0.8532** | **0.8220** | - |

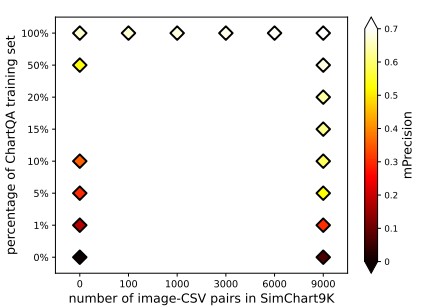

Figure 3: mPrecision by various amounts of charts in ChartQA & SimChart9K.

Table 4: QA results on ChartQA, where LCT and our STR are two different text representation formats. Exact Match (EM) metric is employed for evaluation.

| | Model | Train Set | ChartQA val | | |
|---|---|---|---|---|---|
| | | | aug. | human | avg. |
| Baseline | VL-T5-OCR (Masry et al., 2022) | ChartQA | - | - | 41.6 |
| | Tapas-OCR (Masry et al., 2022) | ChartQA | - | - | 45.5 |
| | PaLI-17B (Chen et al., 2022b) | ChartQA | 64.9 | 30.4 | 47.6 |
| | Pix2Struct (Lee et al., 2022) | ChartQA | 81.6 | 30.5 | 56.0 |
| | MatCha (Liu et al., 2022b) | ChartQA/PlotQA/C.D. | 90.2 | 38.2 | 64.2 |
| | Deplot (Liu et al., 2022a)+GPT3.5 | ChartQA/PlotQA/C.D. | 69.3 | 36.6 | 52.9 |
| LCT | StructChart+GPT3.5 | ChartQA | 64.2 | 37.1 | 50.7 |
| | | Raal Merging | 69.9 | 39.1 | 54.5 |
| | | ChartQA0.2+SimChart9K | 62.6 | 36.0 | 49.3 |
| | | ChartQA+SimChart9K | 71.3 | 41.2 | 56.3 |
| STR | StructChart+GPT3.5 | ChartQA | 78.5 | 42.8 | 60.7 |
| | | Raal Merging | 80.2 | 44.3 | 62.2 |
| | | ChartQA0.2+SimChart9K | 76.3 | 40.7 | 58.5 |
| | | ChartQA+SimChart9K | **83.9** | **46.7** | **65.3** |

related with the number of real-world training samples. (2) Training only on the simulated dataset without any real samples (zero-shot training) fails to achieve a satisfactory CIE performance, due to the insufficiency in real-world charts. (3) By leveraging the proposed method to generate many simulated charts (SimChart9K), only 20% real-world charts can basically achieve equal CIE performance under the 100% real-world training samples. Besides, it can be observed from Table 5 that the CIE performance obtained using 50% real-world charts significantly outperforms that obtained using the full-set training examples. Besides, we conduct experiments on PloatQA and Chart2Text, and the results in Table 6 show that, with the help of SimChart9K, only 10% original real samples in PlotQA and 20% in Chart2Text can achieve equivalent CIE performance under the 100% real training samples. We further illustrate the above observations in Fig. 3.

## 4.4 FROM PERCEPTION TO REASONING: STRUCTURED TRIPLET REPRESENTATIONS (STR)

We conduct experiments to verify the effectiveness of STR as the intermediate representations of chart for various downstream reasoning tasks. We compare STR with commonly-used LCT. Besides, we leverage GPT-3.5 (Brown et al., 2020) as the reasoning module in a one-shot prompting way.

**For QA Task**, we evaluate on ChartQA (Masry et al., 2022) using the Exact Match (EM) metric for the text answer, where a 5% tolerance for the numerical answer is allowed to make a fair comparison. Table 4 shows that: (1) SCRM can effectively reflect the quality of CIE, as verified on the QA task, (2) By comparing LCT, the proposed STR facilitates a better understanding of LLMs, yielding higher answer accuracies. In our analysis, this is mainly due to that the STR is designed using a structural description, with a row-column matching relation compared with the previous LCT format. (3) Our StructChart+GPT3.5 pipeline surpasses all baselines on the ChartQA validation set and achieves comparable QA performance with the recently-proposed method (Liu et al., 2022b) trained on a closed-source dataset (actually covers a large number of charts in the wild). We also verify the performance of QA task on FigureQA (Kahou et al., 2017) in Table 7.

Table 5: Zero-shot and few-shot on real-world chart dataset by means of the proposed SimChart9K.

| | Train Set | $IoU_{thr} \rightarrow$ | mPrecision | Precision | | | |
|---|---|---|---|---|---|---|---|
| | | Tolerance ↓ | 0.5:0.05:0.95 | 0.5 | 0.75 | 0.95 | 1 (EM) |
| No Simulated Source | ChartQA0.01 | strict ■ | 0.1797 | 0.2121 | 0.1771 | 0.1591 | 0.1534 |
| | | slight ■ | 0.2384 | 0.2850 | 0.2339 | 0.2178 | - |
| | | high ■ | 0.2686 | 0.3277 | 0.2633 | 0.2424 | - |
| | ChartQA0.1 | strict ■ | 0.3616 | 0.4015 | 0.3598 | 0.3277 | 0.3210 |
| | | slight ■ | 0.4242 | 0.4612 | 0.4195 | 0.4025 | - |
| | | high ■ | 0.4653 | 0.5038 | 0.4631 | 0.4422 | - |
| | ChartQA0.5 | strict ■ | 0.5389 | 0.5786 | 0.5350 | 0.5085 | 0.4981 |
| | | slight ■ | 0.6150 | 0.6525 | 0.6089 | 0.5890 | - |
| | | high ■ | 0.6603 | 0.6951 | 0.6553 | 0.6392 | - |
| | ChartQA | strict ■ | 0.6770 | 0.7273 | 0.6714 | 0.6458 | 0.6326 |
| | | slight ■ | 0.7792 | 0.8220 | 0.7746 | 0.7519 | - |
| | | high ■ | 0.8274 | 0.8703 | 0.8210 | 0.8011 | - |
| Zero-shot | SimChart9K | strict ■ | 0.0688 | 0.0890 | 0.0691 | 0.0483 | 0.0483 |
| | | slight ■ | 0.1577 | 0.1979 | 0.1562 | 0.1288 | - |
| | | high ■ | 0.2527 | 0.3021 | 0.2491 | 0.2235 | - |
| Few-shot | ChartQA0.01+SimChart9K | strict ■ | 0.3074 | 0.3797 | 0.3078 | 0.2500 | 0.2348 |
| | | slight ■ | 0.4672 | 0.5227 | 0.4706 | 0.4138 | - |
| | | high ■ | 0.5402 | 0.5909 | 0.5398 | 0.4981 | - |
| | ChartQA0.1+SimChart9K | strict ■ | 0.5973 | 0.6619 | 0.6013 | 0.5483 | 0.5294 |
| | | slight ■ | 0.7466 | 0.7936 | 0.7500 | 0.7112 | - |
| | | high ■ | 0.7980 | 0.8419 | 0.8002 | 0.7661 | - |
| | ChartQA0.2+SimChart9K | strict ■ | 0.6465 | 0.7055 | 0.6468 | 0.5956 | 0.5814 |
| | | slight ■ | 0.7787 | 0.8229 | 0.7775 | 0.7443 | - |
| | | high ■ | 0.8206 | 0.8646 | 0.8201 | 0.7888 | - |
| | ChartQA0.5+SimChart9K | strict ■ | 0.6902 | 0.7434 | 0.6913 | 0.6487 | 0.6420 |
| | | slight ■ | 0.8015 | 0.8466 | 0.8002 | 0.7642 | - |
| | | high ■ | 0.8380 | 0.8788 | 0.8400 | 0.8040 | - |

Table 6: Generalizability study for few-shot on PlotQA and Chart2Text by means of SimChart9K.

| Val Set | Model | Train Set | $IoU_{thr} \rightarrow$ | mPrecision | Precision | | | |
|---|---|---|---|---|---|---|---|---|
| | | | Tolerance ↓ | 0.5:0.05:0.95 | 0.5 | 0.75 | 0.95 | 1 (EM) |
| PlotQA | Our StructChart | PlotQA | strict ■ | 0.1995 | 0.2500 | 0.1931 | 0.1765 | 0.1736 |
| | | | slight ■ | 0.7848 | 0.8519 | 0.7784 | 0.7405 | - |
| | | | high ■ | 0.8271 | 0.8922 | 0.8223 | 0.7861 | - |
| | Our StructChart | PlotQA0.1+SimChart9K | strict ■ | 0.1887 | 0.2452 | 0.1874 | 0.1684 | 0.1612 |
| | | | slight ■ | 0.7976 | 0.6624 | 0.7948 | 0.7517 | - |
| | | | high ■ | 0.8063 | 0.8689 | 0.8033 | 0.7614 | - |
| Chart2Text | Our StructChart | Chart2Text | strict ■ | 0.1936 | 0.2473 | 0.1892 | 0.1533 | 0.1442 |
| | | | slight ■ | 0.5524 | 0.6603 | 0.5529 | 0.4672 | - |
| | | | high ■ | 0.6945 | 0.7676 | 0.6934 | 0.6356 | - |
| | Our StructChart | Chart2Text0.2+SimChart9K | strict ■ | 0.2610 | 0.3177 | 0.2601 | 0.2248 | 0.1729 |
| | | | slight ■ | 0.5711 | 0.6731 | 0.5698 | 0.4872 | - |
| | | | high ■ | 0.6871 | 0.7512 | 0.6855 | 0.6309 | - |

**For Summarization and Redrawing Tasks**, due to the lack of public datasets and annotations, it is difficult to provide quantitative results. Thus we present qualitative results by employing different intermediate representations (LCT and STR) in Appendix H, which further shows that the STR has a stronger ability to represent chart information and help chart understanding. More comparisons with other general vision language models (*i.e.* GPT-4V (OpenAI, 2023) and LLaVA-1.5 (Liu et al., 2023)) on downstream tasks are shown in Figs. 13, 14 and 15 of Appendix I.

## 5 CONCLUSION AND OUTLOOK

This work has addressed the task of extracting and understanding the structured information from a visual chart. We propose a plot-to-triplet transformation to achieve objectivity and precision for chart perception. Besides, we leverage the LLM to generate more query data and drawing codes to enhance the generalization ability under practical settings *e.g.* few-shot chart perception, chart redrawing, and question answering. For future work, we may seek an end-to-end framework with our techniques, which is still an open question.

**Ethics Statement.** Our proposed StructChart is useful in many fields for Chart Understanding (CU), such as medical, education, finance, business, *etc*. One potential negative societal impact is: our approach may perform chart image plagiarism through redrawing task, which may raise privacy concerns and result in the academic misconduct. Nevertheless, our model represents chart image by the extracted Structured Triplet Representations (STR). On the positive side, this can be used for chart duplication checking.

**Reproducibility Statement.** We have included the complete schematic simulation process for Sim-Chart9K in Appendix E, and the source code is attached in supplementary material. Furthermore, our simulation chart dataset SimChart9K and checkpoints of StructChart will be released soon.

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

# A  RELATED WORKS ON VISION LANGUAGE PRE-TRAINED MODELS (VLPMs)

Here, we also discuss related works on Vision Language Pre-trained Models (VLPMs). According to the way of aggregating information from different modalities, VLPMs can be categorized into **fusion-encoder based**, **dual-encoder based**, and **combination based** models.

**Fusion-encoder based** VLPMs take text embeddings and image features as input and involve several fusion approaches to model Vision Languate (VL) interaction. VL-BERT (Su et al., 2019), UNIMO (Li et al., 2020; 2022), SOHO (Huang et al., 2021), VL-T5 (Cho et al., 2021), SimVLM (Wang et al., 2021), and ViLT (Kim et al., 2021) assume that the potential correlation and alignment between modalities can be learned by a single transformer encoder. Thus, the text embeddings and image features are concatenated with additional embeddings that indicate position and modalities, and fed into a transformer-based encoder. Further, ViLBERT (Lu et al., 2019), Visual Parsing (Xue et al., 2021), ALBEF (Li et al., 2021a), and WenLan (Huo et al., 2021) adopt a cross-attention mechanism to model the interaction between VL modalities, where the query vectors originate from one modality and the key and value vectors from the other. Typical pre-training tasks for fusion-encoder based VLPMs include: masked language/vision modeling, image-text matching, masked region classification, masked region feature regression, visual grounding, visual question answering, and grounded captioning. Thus, fusion-encoder based VLPMs can be effective in VL understanding downstream tasks.

**Dual-encoder based** VLPMs utilize two individual single-modal encoders to encode each modality separately, then convert the image and text embeddings into the same semantic space to calculate the VL similarity scores. CLIP (Cho et al., 2021), ALIGN (Jia et al., 2021) and DeCLIP (Li et al., 2021b) leverage large-scale image-text pairs to learn transferable visual representations for retrieval tasks and exhibit surprising zero-shot transfer to image classification tasks.

**Combination based** VLPMs combine the benefits of fusion-encoder based and dual-encoder based architectures. FLAVA (Singh et al., 2021) firstly utilizes a dual-encoder to acquire single-modal representations. Then, the single-modal embeddings are processed by a fusion-encoder to obtain cross-modal representations. VLMo (Wang et al., 2022) introduces a novel approach called Mixture-of-Modality Expert (MoME), combining a dual-encoder and a fusion-encoder into one unified framework, which can be fine-tuned on both VL understanding and image-text retrieval tasks.

# B  DETAILED EXPLANATION OF MATCHING PROCESS BETWEEN PREDICTION AND GROUND TRUTH

Prediction results and Ground Truth (GT) are both in the form of STR, which is a set of tuples $(Entity_{r_n}, Entity_{c_m}, Value_{r_n}^{c_m})$. The complete matching process for these two sets is as follows:

As mentioned in 3.2, **we firstly query whether each tuple in GT set has a matching tuple in prediction set.** Specifically, for two Entity strings in each tuple, we reorder them according to the following pre-process: 1) lowercase all characters in Entity strings; 2) the two Entity strings in each Tuple are sorted by the ASCII of the first letter; 3) if the ASCII of the first letter is the same, the second letter is compared, *etc.* After reordering, we concatenate the two Entity strings together and calculate the edit distance according to Eq. 3. Actually, $Entity_{pred}^{p}$ and $Entity_{GT}^{q}$ in Eq. 3 are both concatenated strings after reordering. For the value string, we convert it to floating-point and calculate the relative error according to Eq. 4. When the editing distance and relative error are less than the threshold, the two tuples (the prediction tuple and GT tuple) are matched. It should be noted that under different tolerance level, the threshold of edit distance and relative error are different for performing more comprehensive evaluation.

Then, **we calculate the Intersection over Union ($IoU$) between prediction set and GT set** according to Eq. 7 in the main text. If the calculated $IoU$ is larger than the preset threshold $IoU_{thr}$, the prediction and GT are matched. Also, the preset threshold $IoU_{thr}$ here is variable for more flexible evaluation.

Table 7: QA results on FigureQA. Exact Match (EM) metric is employed for evaluation.

| | Model | Train Set | FigureQA val |
|---|---|---|---|
| **Baseline** | MatCha (Liu et al., 2022b) | ChartQA/PlotQA/C.D. | - |
| | Deplot (Liu et al., 2022a)+GPT-3.5 | ChartQA/PlotQA/C.D. | 53.9 |
| **STR** | StructChart+GPT3.5 | Real&Sim Merging | **63.3** |

## C  EXPERIMENTAL RESULTS AND EMPIRICAL ANALYSES OF MATCHA AND DEPLOT ON QA TASK

Matcha (Liu et al., 2022b) is an end-to-end model (without the help of GPT-3 or 3.5), and we cannot modify its reasoning module (including network design and language model selection) during the evaluation process. As a result, we only choose to directly compare with Matcha. Experimental results in Table 4 show that our StructChart has a 1.1% lead over Matcha (Liu et al., 2022b) in average accuracy.

However, Deplot (Liu et al., 2022a) applies Chain-of-Thought (CoT) (Wei et al., 2022), Program-of-Thought (PoT) (Chen et al., 2022a) and Self-Consistency (SC) (Fan et al., 2022) strategies to help the reasoning module boost the performance. Since we cannot access specific prompt designs in Deplot (Liu et al., 2022a), the fairness of result comparison on QA task cannot be guaranteed.

For fair comparison, we evaluate StructChart and Deplot (Liu et al., 2022a) with the same reasoning model (GPT-3.5) setting without any prompt engineering strategies nor evaluation tactics. As reported in Table 4 in the main text, our StructChart has a 12.4% lead over Deplot for QA task on ChartQA.

## D  QA RESULTS ON FIGUREQA BY STRUCTCHART AND OTHER METHODS

FigureQA (Kahou et al., 2017) is also a benchmark for chart understanding task. However, there are still some differences between FigureQA and the mentioned datasets (ChartQA, PlotQA and Chart2Text): 1) Overall, the images in FigureQA dataset are all simulated samples based on manual rules, while ChartQA and Chart2Text are sampled from real world. 2) For perception stage, ChartQA and Chart2Text provide annotations in CSV format for CIE task. But for FigureQA and PlotQA, we need to preprocess relatively complex retrieval from the raw annotation (JSON file). 3) For reasoning stage, the questions in FigureQA dataset are based on fifteen sentence patterns (e.g. "Is X the minimum?", "Does X have the lowest value?"). And all the answer pairs are "yes" or "no". In contrast, the mentioned datasets, such as ChartQA, PlotQA, use human-authored QA annotations. Besides, Chart2Text is designed for chart summarization task.

## E  LLM-BASED SELF-INSPECTION DATA PRODUCTION SCHEME

The complete schematic simulation process can be divided into two stage, containing (1) Data simulation stage for label generation. (2) Image simulation stage for chart generation. We demonstrate two examples in Figs. 4 and 5.

## F  VISUALIZATION COMPARISONS WITH MATCHA AND DEPLOT

We compare the proposed Structchart with Matcha (Liu et al., 2022b) and Deplot (Liu et al., 2022a) in Chart Information Extraction (CIE) task in Figs. 6, 7 and 8. Matcha (Liu et al., 2022b) cannot perform CIE task and can only output the content in the form of HTML format during the pretraining phase, and Deplot (Liu et al., 2022a) is affected by noisy backgrounds (*e.g.* web page environment containing other irrelevant text). By comparison, our Structchart can extract accurate information of the chart from complicated background, such as information from the website.

| Data Simulation Stage |
|---|
| **Prompt:** |
| Copying the following table information can be expanded and adapted as appropriate, The imitation is as irrelevant as possible to the original text. |
| **Input LCT data:** |
| Year,It matters wins the really who presidential election,Things will be pretty much the same regardless of who is elected/n 2000,50,44/n 2004,67,29/n 2008,63,32/n 2012,63,34/n 2016,74,22/n 2020,83,16 |
| **Simulated LCT data (completed by GPT-3.5):** |
| Year,People would be better off if the government would just stay out of it,People would be worse off if the government did not get involved/n 2001,45,48/n 2005,53,42/n 2009,60,36/n 2013,63,31/n 2017,71,25/n 2021,76,19 |
| Image Simulation Stage |
| **Prompt:** |
| Consider you are a professional Python grapher. Please draw and save a chart based on the following data using python, and images must be clear and intuitive. Choose a plot type that best suits the value, for example line, column, scatter and pie charts. Drawing techniques such as background grids can be used. |
| ...... |
| **Code generated by GPT3.5:** |

```
import matplotlib.pyplot as plt
data = [[2001, 45, 48],[2005, 53, 42],[2009, 6, 36],[2013, 63, 31],[2017, 71, 25],[2021, 76, 19]] year,
better_off,worse_off = zip(*data)
fig,ax = plt.subplots(figsize=(15,7))
ax.plot(year,better off,color='green",label-'People would be better off if the government would just stay
out of it')
ax.plot(year.worse off, color='red', label='people would be worse off if the government did not get in-
volved')
ax.legend(loc='upper left",fontsize=12,ncol=2)
ax.set_xlabel('Year',fontsize=14)
ax.set_ylabel("Number',fontsize=14)
ax.set_title('Government Involvement', fontsize=14fontweight='bold')
ax.set_xticks(year)
plt.tight_layout()
plt.savefig("34.png",dpi-400)
plt.cla()
```

**Chart image simulated:**

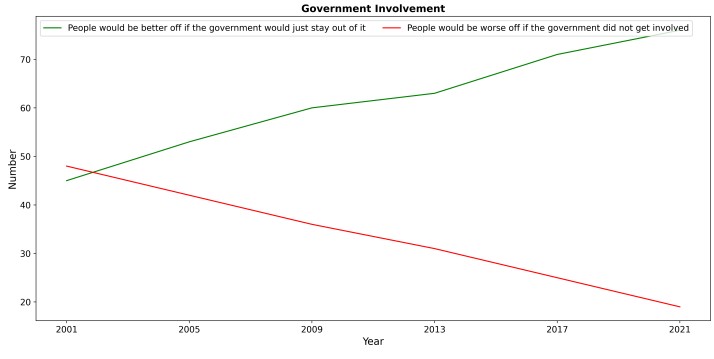

Figure 4: Example One of label-efficient chart simulation process including: (1) data simulation stage for labels generation. (2) image simulation stage for charts generation.

| Data Simulation Stage |
|---|

**Prompt:**
Copying the following table information can be expanded and adapted as appropriate. The imitation is as irrelevant as possible to the original text.

**Input LCT data:**
Entity,Value/n
Have not heard of Latinx,76.0/n
Have not heard of Latinx,76.0/n
Do not use Latinx,20.0/n
Do not Latinx u,20.0/n

**Simulated LCT data (completed by GPT-3.5):**
Characteristic, Share of engineers/n
Very satisfied,48% /n
Satisfied,41% /n
Neither satisfied nor dissatisfied,7% /n
Dissatisfied,3% /n
Very dissatisfied,1% /n

| Image Simulation Stage |
|---|

**Prompt:**
Consider you are a professional Python grapher.
Please draw and save a chart based on the following data using python, and images must be clear and intuitive.
Choose a plot type that best suits the value, for example, line, column, scatter and pie charts.
Drawing techniques such as background grids can be used.

......

**Code generated by GPT3.5:**
```python
import matplotlib.pyplot as plt
Char= ['Very Satisfied', 'Satisfied', 'Neither Satisfied not Dissatisfied', 'Dissatisfied', 'Very Dissatisfied']
Share=[48, 41, 7, 3, 1]
plt.figure(figsize=(5,5))
plt.pie(Share,labels=Char,autopct='%1.2f%%',textprops='fontsize': 10)
plt.title('EngineersŚatisfaction Level')
plt.tight_layout()
plt.savefig('86.png')
plt.cla()
```

**Chart image simulated:**

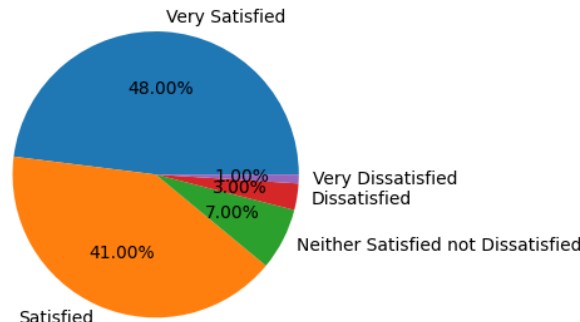

Figure 5: Example Two of label-efficient chart simulation process including: (1) data simulation stage for labels generation. (2) image simulation stage for charts generation.

## G  MORE RESULTS OF STRUCTCHART ON DIFFERENT DOWNSTREAM TASKS

We visually demonstrate StructChart on downstream tasks in Figs. 9 and 10, including Question Answering (QA), summarization, and redrawing. For QA task, quantitative evaluation results are shown in Tab 4, and here, we further give many visualization results. For summarization task, some open-ended summary descriptions can be conducted beyond the basic numeric description. For redrawing, different types of charts can be obtained by redrawing chart image given the statistical data (*e.g.*, line chart → bar chart, bar chart → pie chart, *etc.*)

## H  DEMONSTRATIONS ON DOWNSTREAM TASKS WITH LINEAR CSV TOKENS (LCT) V.S. STRUCTURED TRIPLET REPRESENTATIONS (STR)

We respectively use Linear CSV Tokens (LCT) and Structured Triplet Representations (STR) as intermediate representations of chart information for different downstream tasks. Figs. 11 and 12 show that STR used in StructChart has better robustness compared to LCT. When noise is introduced into the highly position-sensitive LCT (a comma is introduced as noise as illustrated in Fig. 11, and separator comma itself is included as illustrated in Fig. 12), all downstream tasks will be affected negatively. By comparison, our StructChart achieves better performance on QA, summarization, and redrawing tasks, owing to the proposed STR.

## I  COMPARISONS WITH OTHER GENERAL VISION LANGUAGE MODELS (VLM) ON DOWNSTREAM TASKS

We demonstrate three visualization comparisons among StrcutChart, GPT-4V (OpenAI, 2023) and LLaVA-1.5 (Liu et al., 2023) for downstream tasks in Figs. 13, 14 and 15. It is worth noting that for QA task, our StructChart is restricted to generating answers only in order to report the quantitative results on ChartQA dataset. From the visualization results, it can be seen that the proposed StructChart has a strong Chart-based reasoning ability, showing promising performance on multiple downstream tasks. Although GPT-4V (OpenAI, 2023) also has a strong performance, our model weights, training code, simulation code, and simulation dataset will all be open-sourced.

## J  DISCUSSION ABOUT HOW TO OBTAIN GENERAL CHART LARGE MODEL (CLM)

Here, we discuss one possible direction for training a Chart Large Model (CLM), which is challenging due to: (1) Scarce chart data covering as comprehensive scientific fields as possible; (2) Perception-reasoning task gap. This paper provides a preliminary attempt to tackle the above challenges. In the future, we intend to collect more chart data from different scientific subjects and perform the proposed LLM-based self-inspection data production scheme to enhance chart data diversity. This would be the foundation for training a general CLM with a large amount of simulated chart data which are rendered using real-world chart data from multiple fields.

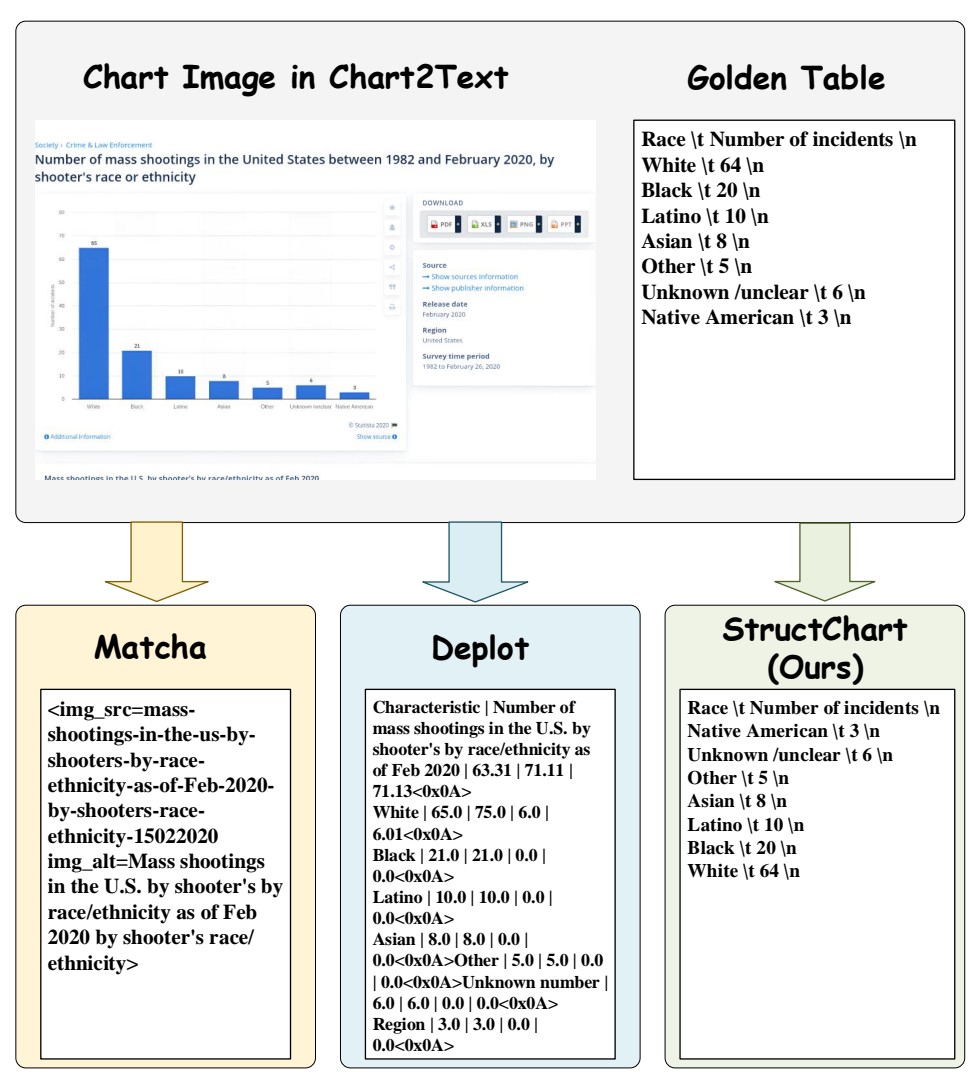

Figure 6: Comparison of the proposed StructChart, Matcha (Liu et al., 2022b) and Deplot (Liu et al., 2022a), where the Golden Table represents the ground truth of the parsed chart information.

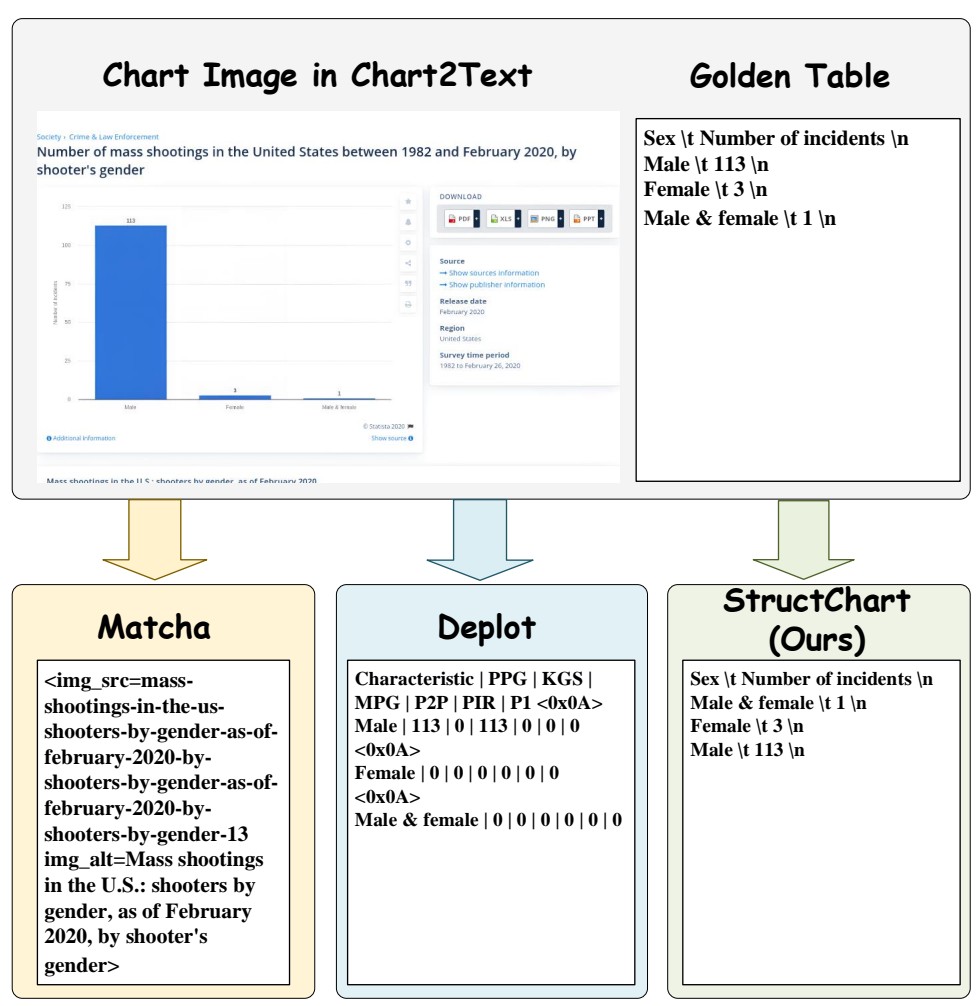

Figure 7: Comparison of the proposed StructChart, Matcha (Liu et al., 2022b) and Deplot (Liu et al., 2022a), where the Golden Table represents the ground truth of the parsed chart information.

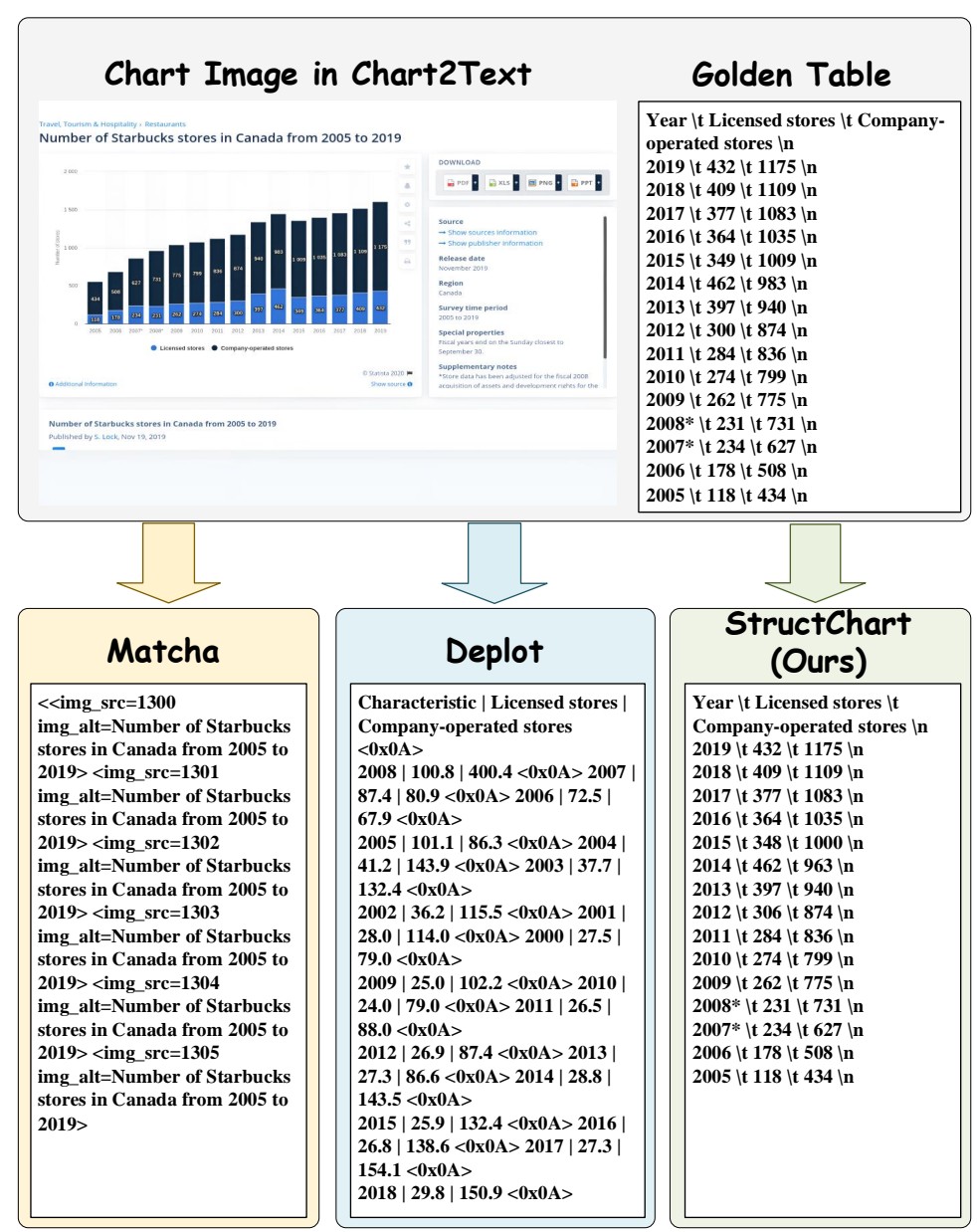

Figure 8: Comparison of the proposed StructChart, Matcha (Liu et al., 2022b) and Deplot (Liu et al., 2022a), where the Golden Table represents the ground truth of the parsed chart information.

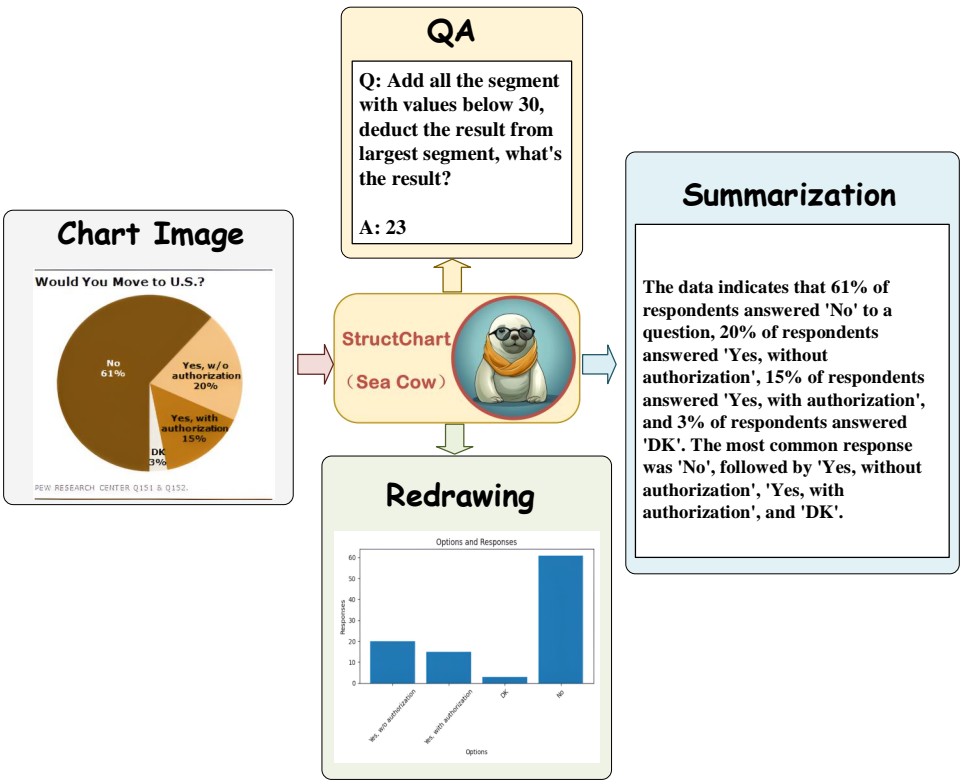

Figure 9: Visualization results using the proposed StructChart on different chart-related reasoning tasks including Question Answering (QA), Summarization, and Redrawing.

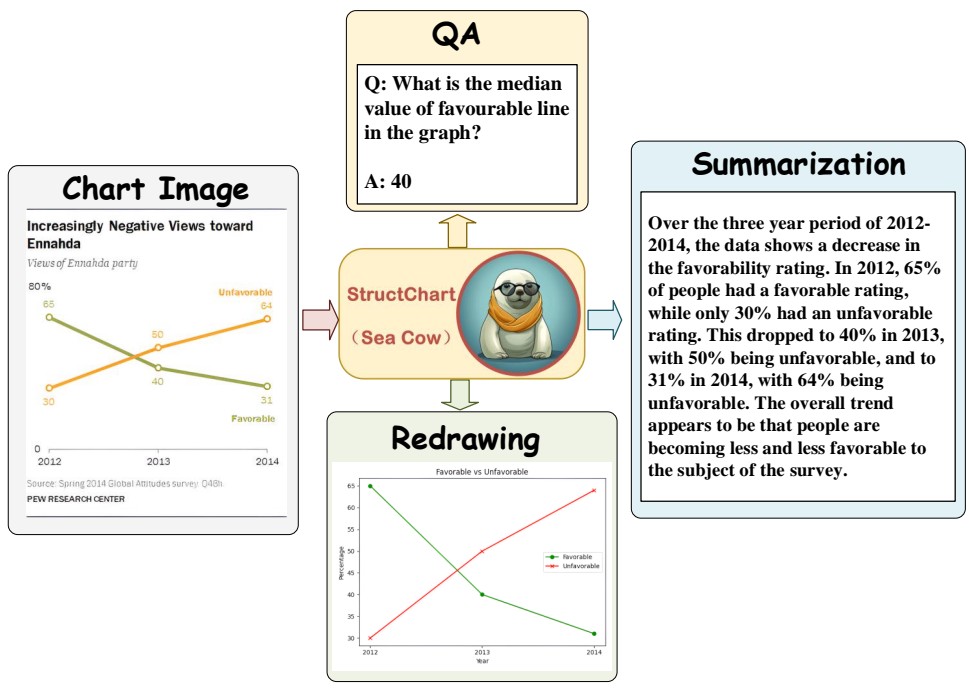

Figure 10: Visualization results using the proposed StructChart on different chart-related reasoning tasks including Question Answering (QA), Summarization, and Redrawing.

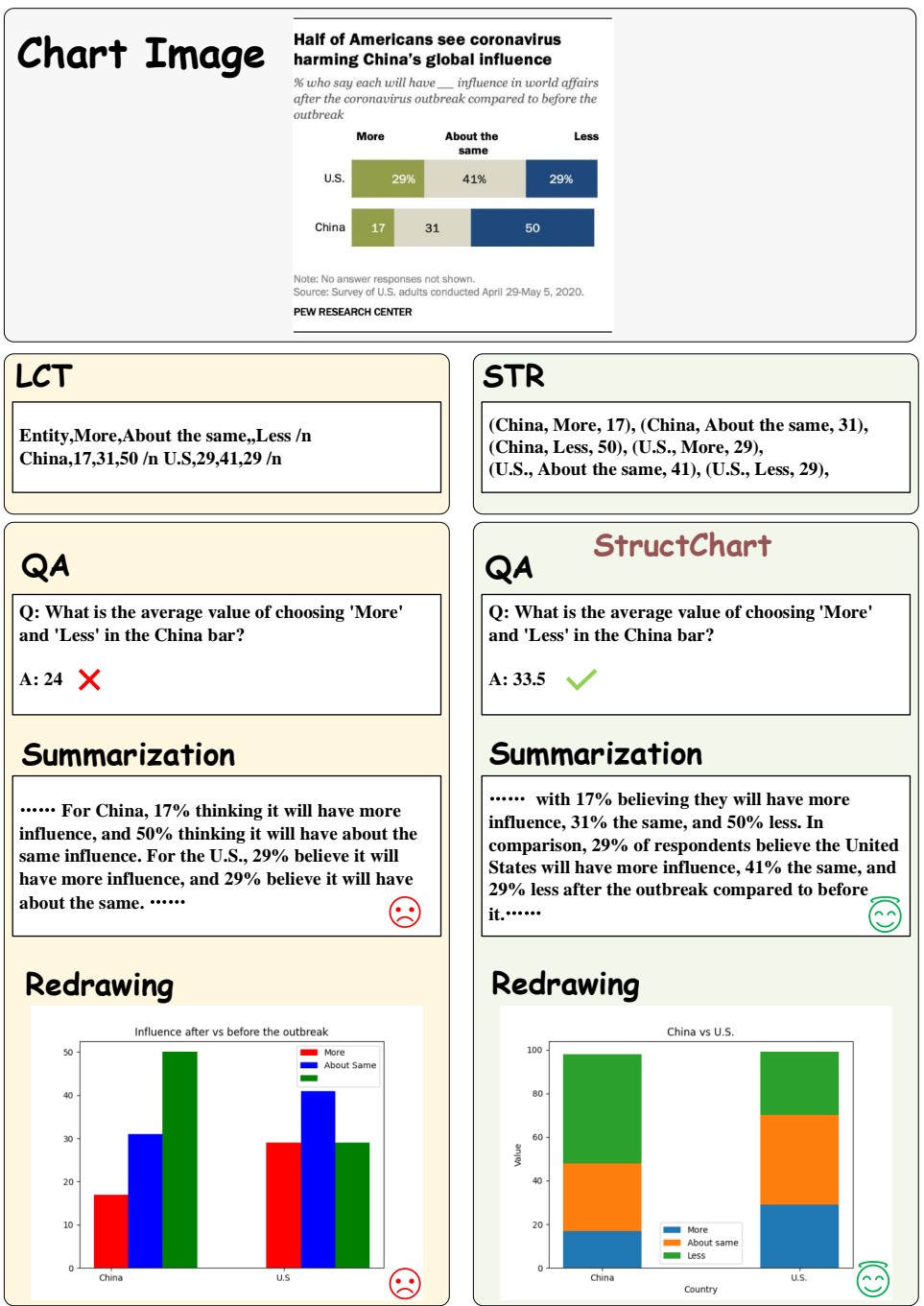

Figure 11: Visualization results (a comma introduced in LCT) using the proposed StructChart on downstream tasks with Linear CSV Tokens (LCT) v.s. Structured Triplet Representations (STR).

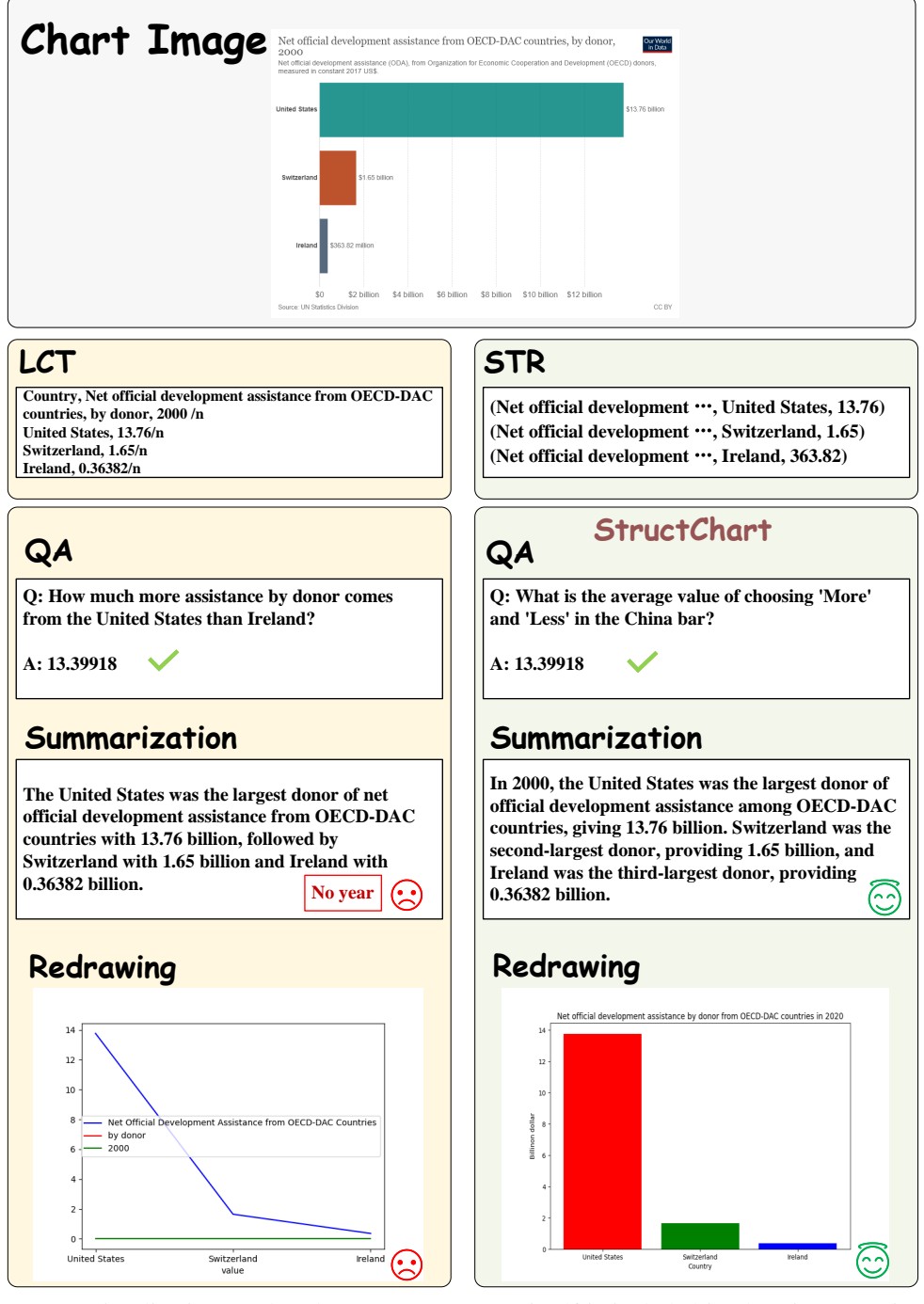

Figure 12: Visualization results (the separator comma itself is included in chart image) using the proposed StructChart on downstream tasks with Linear CSV Tokens (LCT) v.s. Structured Triplet Representations (STR).

## Chart Image

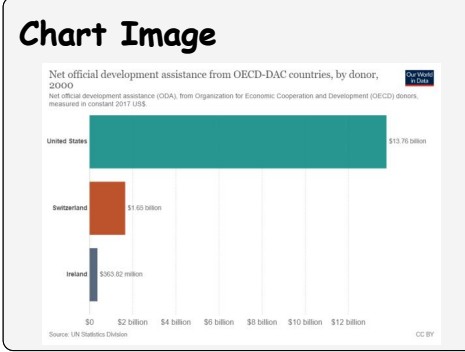

Net official development assistance from OECD-DAC countries, by donor, 2000
Net official development assistance (ODA), from Organization for Economic Cooperation and Development (OECD) donors, measured in constant 2017 US$.

## StructChart

### QA

Q: How much more assistance by donor comes from the United States than Ireland?

A: 13.39618 ✓

### Summarization

In 2000, the United States was the largest donor of official development assistance among OECD-DAC countries, giving 13.76 billion. Switzerland was the second-largest donor, providing 1.65 billion, and Ireland was the third-largest donor, providing 0.36382 billion.

### Redrawing

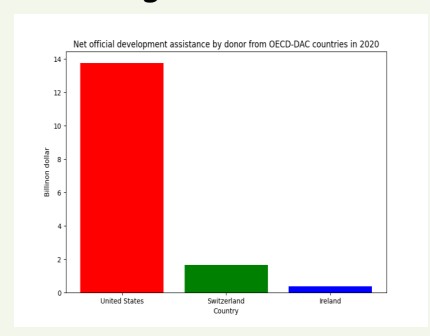

## LLAVA-1.5

### QA

Q: How much more assistance by donor comes from the United States than Ireland?

A: the United States is shown to provide $36.5 billion in assistance, while Ireland provides only $1.2 billion. This indicates that the United States is a much larger donor in terms of financial assistance to other countries compared to Ireland. ✗

### Summarization

The image shows a comparison of the amount of assistance provided by donor countries to other countries. The United States is the largest donor, providing $36.5 billion in assistance, while Ireland provides only $1.2 billion. This highlights the difference in the scale of assistance provided by these two countries to other nations.

## GPT-4V

### QA

Q: How much more assistance by donor comes from the United States than Ireland?

S: The chart shows that the United States provided $13.76 billion while Ireland provided $363.82 million. We can subtract Ireland's contribution from the United States' contribution.
$13,760,000,000 - $363,820,000 = $13,396,180

Answer: $13,396,180 ✓

### Summarization

The chart shows net official development assistance (ODA) for 2000 from three OECD-DAC countries. The United States leads with $13.76 billion, Switzerland provided $1.65 billion, and Ireland contributed $363.82 million, all in constant 2017 dollars.

Figure 13: Visualization comparisons among StructChart, LLaVA-1.5 (Liu et al., 2023) and GPT-4V (OpenAI, 2023) on downstream tasks.

## Chart Image

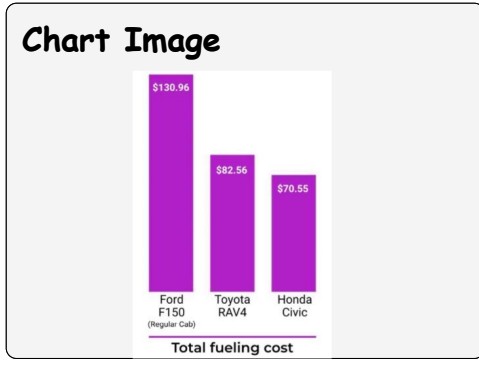

## StructChart

### QA

Q: What is the average total fueling cost excluding the Ford F150?

A: 76.55 ✓

### Summarization

The data shows that the Ford F150 (Regular Cab) has the highest cost of fueling at $130.96, followed by Toyota RAVA at $82.56 and Honda Civic at the lowest cost of $70.55.

### Redrawing

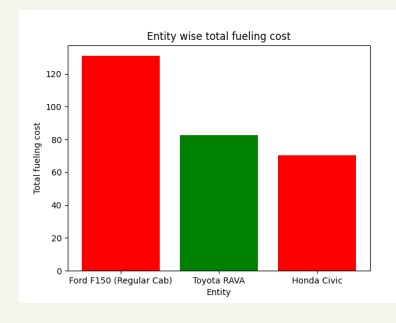

## LLAVA-1.5

### QA

Q: What is the average total fueling cost excluding the Ford F150?

A: The average total fueling cost excluding the Ford F150 is $7.50. ✗

### Summarization

The image displays a comparison of the total fueling cost for various vehicles, including a Ford F150, a Toyota RAV4, and a Honda Civic. The chart shows the cost of fueling each vehicle, with the Ford F150 being the most expensive at $13.99, followed by the Toyota RAV4 at $7.50, and the Honda Civic at $7.00. The chart is presented in a purple and white color scheme, with the vehicle names and fueling costs clearly visible.

## GPT-4V

### QA

Q: What is the average total fueling cost excluding the Ford F150?

S: To find the average total fueling cost excluding the Ford F150, we need to add the total fueling costs of theToyota RAV4 and Honda Civic and divide by 2, (82.56 + 70.55)+ 2 = 76.55. Therefore, the Answer is $76.55

Answer: $76.55 ✓

### Summarization

The bar chart compares the total fueling cost of three different vehicles, showing the Ford F150 with the highest cost at $130.96, followed by the Toyota RAV4 at $82.56, and the Honda Civic with the lowest at $70.55.

Figure 14: Visualization comparisons among StructChart, LLaVA-1.5 (Liu et al., 2023) and GPT-4V (OpenAI, 2023) on downstream tasks.

## Chart Image

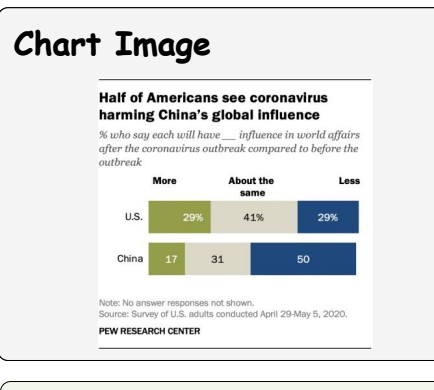

### StructChart

**QA**

Q: What is the average value of choosing 'More' and 'Less' in the China bar?

A: 33.5 ✓

### Summarization

······ with 17% believing they will have more influence, 31% the same, and 50% less. In comparison, 29% of respondents believe the United States will have more influence, 41% the same, and 29% less after the outbreak compared to before it.······

### Redrawing

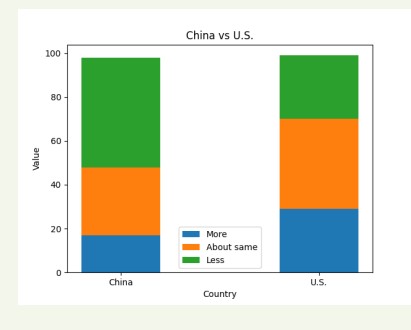

## LLAVA-1.5

**QA**

Q: What is the average value of choosing 'More' and 'Less' in the China bar?

A: The average value of choosing 'More' and 'Less' in the China bar is 40%. ✗

### Summarization

The image features a chart that shows the percentage of Americans who see the coronavirus as harming China's global influence. The chart is divided into three sections, with the top section showing the percentage of people who believe the virus is harming China's influence, the middle section showing the percentage who believe it is not harming China's influence, and the bottom section showing the percentage who are unsure.

## GPT-4V

**QA**

Q: What is the average value of choosing 'More' and 'Less' in the China bar?

S: To calculate the average value of the percentages for "More" and "Less" for China, you would add the two percentages together and then divide by two. (17 + 50) / 2 = 67 / 2 = 33.5%

Answer: 33.5% ✓

### Summarization

A Pew Research Center survey shows that half of Americans believe the coronavirus will decrease China's global influence. While 29% think the U.S.'s influence will increase, 50% feel China's will diminish post-outbreak. This survey reflects U.S. perceptions during April 29-May 5, 2020.

Figure 15: Visualization comparisons among StructChart, LLaVA-1.5 (Liu et al., 2023) and GPT-4V (OpenAI, 2023) on downstream tasks.

