# OpenReview forum: "StructChart: Perception, Structuring, Reasoning for Visual Chart Understanding"
_ICLR.cc/2024/Conference — Submitted to ICLR 2024_

### Official Review · Reviewer_Rkaq · 2023-10-30

**Soundness:** 3 good
**Presentation:** 3 good
**Contribution:** 4 excellent
**Rating:** 6
**Confidence:** 5

**Summary:**

The authors claim that they proposed a unified and label-efficient learning paradigm for joint perception and reasoning tasks, which can be generally applicable to different downstream tasks, beyond the question-answering task as specifically studied in peer works.

**Strengths:**

The authors claim that they proposed a unified and label-efficient learning paradigm for joint perception and reasoning tasks, which can be generally applicable to different downstream tasks, beyond the question-answering task as specifically studied in peer works.

**Weaknesses:**

1. The experiments should focus on the generalizability of the conclusions/findings derived from the powerful transformer-based models, and it remains a concern.

2. What is the relation and difference with the current popular benchmark (e.g., FigureQA[1])?

[1] Kahou, Samira Ebrahimi, et al. "Figureqa: An annotated figure dataset for visual reasoning." arXiv preprint arXiv:1710.07300 (2017).

**Questions:**

Please refer to Weakness.

---

> ### Author Response · Authors · 2023-11-16
> **Response to Reviewer Rkaq: Part 1/2**
>
> **Q1: The experiments should focus on the generalizability of the conclusions/findings derived from the powerful transformer-based models, and it remains a concern.**
>
> **A1:** Thank you for providing such constructive comments! It also gets us to rethink the generalizability of the conclusions/findings. In this regard, we use ChartQA, PlotQA and Chart2Text as the validation sets for the perception stage, and carry out supplementary experiments on the following conclusions to verify the generalization ability from the following two aspects:
>
> 1. In Section 4.2 of the main text, we claim that, larger scale of the data will lead to better performance for CIE task, both in real data domain and our simulated domain. The following experimental results show that, the larger the amount of real-world or simulation data, the better the CIE task performance on **all** the ChartQA, PlotQA, and Chart2Text datasets. "Merging" in the following tables refers to that, training samples are merged from ChartQ, PlotQA, and Chart2Text.
>
> | Val Set | Model       | Train Set          | mPrecision (strict) |  mPrecision (slight)      |   mPrecision (high)     | Precision(EM)|
> |:-------:|:-----------:|------------------|:-------:|:--------:|:--------:|:--------:|
> | ChartQA | StructChart | ChartQA             | 0.6770    | 0.7792 | 0.8274 | 0.6326 |
> | ChartQA | StructChart | Merging            | 0.7017     | 0.8227 | 0.8591 | 0.6506 |
> | ChartQA | StructChart | Merging+SimChart9K | 0.7187     | 0.8311 | 0.8572 | 0.6642 |
>
>
> | Val Set | Model       | Train Set          | mPrecision (strict) |  mPrecision (slight)      |   mPrecision (high)     | Precision(EM)|
> |:-------:|:-----------:|------------------|:-------:|:--------:|:--------:|:--------:|
> | PlotQA | StructChart | PlotQA    |  0.1995  | 0.7848 | 0.8271 | 0.1736 |
> | PlotQA | StructChart | Merging            | 0.4549     | 0.8589 | 0.8921 | 0.3385 |
> | PlotQA | StructChart | Merging+SimChart9K | 0.4596     | 0.8612 | 0.8998 | 0.3612 |
>
>
> | Val Set | Model       | Train Set          | mPrecision (strict) |  mPrecision (slight)      |   mPrecision (high)     | Precision(EM)|
> |:-------:|:-----------:|------------------|:-------:|:--------:|:--------:|:--------:|
> | Chart2Text | StructChart | Chart2Text | 0.1936 | 0.5524 | 0.6945 | 0.1442 |
> | Chart2Text | StructChart | Merging            | 0.3156     | 0.7141 | 0.8085 | 0.2381 |
> | Chart2Text | StructChart | Merging+SimChart9K | 0.3394     | 0.7451 | 0.8296 | 0.2635 |
>
>
> 2. In Section 4.3 of the main text, we claim that "We can achieve a high-performance CIE only leveraging few-shot real samples", which has been verified on ChartQA dataset in Table 5 of the main text.  **To further verify the generalizability of this finding**, we also supplement validation experiments with PloatQA and Chart2Text. The results in the following table show that, with the help of SimChart9K, only 10% original real samples in PlotQA and 20% in Chart2Text can basically achieve equivalent CIE performance under the 100% real training samples.
>
> | Val Set | Model       | Train Set          | mPrecision (strict) |  mPrecision (slight)      |   mPrecision (high)     | precision(EM)|
> |:-------:|:-----------:|------------------|-------|--------|--------|--------|
> | PlotQA | StructChart | PlotQA    |  0.1995  | 0.7848 | 0.8271 | 0.1736 |
> | PlotQA | StructChart | PlotQA0.1+SimChart9K |   0.1887  | 0.7976 | 0.8063 | 0.1612 |
> | Chart2Text | StructChart | Chart2Text | 0.1936 | 0.5524 | 0.6945 | 0.1442 |
> | Chart2Text | StructChart | Chart2Text0.2+SimChart9K |   0.2610 |  0.5711 |  0.6871 |  0.1729 |

---

> ### Author Response · Authors · 2023-11-16
> **Response to Reviewer Rkaq: Part 2/2**
>
> **Q2: What is the relation and difference with the current popular benchmark (e.g., FigureQA)?**
>
> **A2:** Thank the reviewer very much for this valuable comment. FigureQA is also a benchmark for chart understanding task. However, there are still some differences between FigureQA and the mentioned datasets (ChartQA, PlotQA and Chart2Text):
> - Overall, the images in FigureQA dataset are all simulated samples based on manual rules, while ChartQA and Chart2Text are sampled from real world.
> - For perception stage, ChartQA and Chart2Text provide annotations in CSV format for CIE task. But for FigureQA and PlotQA, we need to preprocess relatively complex retrieval from the raw annotation (JSON file).
> - For reasoning stage, the questions in FigureQA dataset are based on fifteen sentence patterns (e.g.  "Is X the minimum?", "Does X have the lowest value?"). And all the answer pairs are "yes" or "no". In contrast, the mentioned datasets, such as ChartQA, PlotQA, use human-authored QA annotations. Besides, Chart2Text is designed for chart summarization task.
>
> Furthermore, according to the reviewer's comments, we have supplemented additional experiments for QA task on FigureQA. The output of Matcha is still HTML format (e.g. '('title | title <0x0a> yaxis_label | xaxis_label <0x0a> burlywood | 64 <0x0a> mint | 19.77 <0x0a> dim gray |...'), so the quantitative results of Matcha are almost close to zero. In contrast, our StructChart still works well on FigureQA dataset. For example, our StructChart outperforms Deplot by 9.4% for QA task.
>
> |  Input Representations   | Model   | Train Set  | Acc on FigureQA |
> |:-----:|---------------------|:---------------------:|:------:|
> | \ | Deplot+GPT-3.5 | ChartQA+PlotQA+C.D.  | 53.9 |
> | STR | StructChart+GPT-3.5 | Merging+SimChart9k  | **63.3** |

---

> > ### Comment · Reviewer_Rkaq · 2023-11-22
> >
> > The author has answered my question, but I don't find the relevant sentence in the revised version. Therefore I keep my rating.

---

> > > ### Author Response · Authors · 2023-11-22
> > > **Thanks for your approval, we will update the revised version as soon as possibile**
> > >
> > > Thanks for your response and approval.
> > > The revised manuscript needs to add many experimental tables, and it is still in the process of being modified. These adjustments will be updated into the revised manuscript before the deadline.

---

> > > ### Author Response · Authors · 2023-11-22
> > > **Paper Update**
> > >
> > > Dear Reviewer Rkaq，
> > >
> > > Thanks a lot for your valuable comments, and constructive suggestions, and we have updated the revised paper.
> > >
> > > FYI, we provide a summary of the updates inspired by you:
> > >
> > > - We emphasize the generalizability of the findings (a larger scale of the data will lead to better performance for CIE task) in **Sec 4.2, Page 6** of the main text. Also, we supplement the corresponding generalizability experiments in **Table 2, Page 7** of the main text.
> > >
> > > - We emphasize the generalizability of the findings (“we can achieve a high-performance CIE only leveraging few-shot real samples”) in **Sec 4.3, Page 8** of the main text. Also, we supplement the corresponding generalizability experiments in **Table 6, Page 9** of the main text.
> > >
> > > - For benchmark FigireQA, we provide a brief description in **Sec 4.1, Page 6** of the main text. Meanwhile, we compare FigureQA among other datasets we adopted in this work in **Appendix D, Page 14**. Also, the QA results in FigureQA are shown in **Table 7, Page 14** of the Appendix.
> > >
> > > We all think that your insightful suggestions make our paper more complete and convicing. Sincerely, hoping you reconsider your rating with the revised version.
> > >
> > > Best regards,
> > >
> > > Authors of Paper 1125

---

> > > ### Author Response · Authors · 2023-11-23
> > > **Manuscript has been updated**
> > >
> > > Dear Reviewer Rkaq，
> > >
> > > According to your suggestion during the previous round, we have further revised our manuscript from the following aspects:
> > >
> > > - In Sec 4.2 of Page 6 and Table 2 of Page 7 of the main text, we further supplement the corresponding generalizability experiments.
> > >
> > > - We newly add Table 6 in Page 9 to emphasize the generalizability of the findings (“we can achieve a high-performance CIE only leveraging few-shot real samples”).
> > >
> > > - For benchmark FigireQA, we provide a brief description in Sec 4.1, Page 6 of the main text. Meanwhile, we compare FigureQA among other datasets we adopted in this work in Appendix D, Page 14. Also, the QA results in FigureQA are shown in Table 7, Page 14 of the Appendix.
> > >
> > > **ALL answers during the rebuttal period have been added into the revised manuscript**. It is appreciated that you could consider these modifications and rapid actions of our manuscript.
> > >
> > >
> > > &ensp;
> > >
> > > Best regards,
> > >
> > > Authors of Paper 1125

---

### Official Review · Reviewer_ojHH · 2023-10-31

**Soundness:** 3 good
**Presentation:** 3 good
**Contribution:** 3 good
**Rating:** 6
**Confidence:** 4

**Summary:**

This paper introduces a novel approach known as the integrated perception and reasoning paradigm, designed to enhance the comprehension of visual charts. Initially, StructChart transforms chart data from the common tabular format, such as linearized CSV, into the newly introduced Structured Triplet Representations (STR). Concurrently, we present the Structuring Chart-oriented Representation Metric (SCRM) for a quantitative assessment of performance. Additionally, we investigate the potential of utilizing the capabilities of a Large Language Model (LLM) to expand the training dataset, specifically the SimChart9K dataset.

**Strengths:**

1) The paper introduces a new Structured Triplet Representation (STR) instead of CSV format.
2) The Structuring Chart-oriented Representation Metric (SCRM) is suitable for various tasks related to chart perception.
3) This paper provides the SimChart9K dataset, which leverages Large Language Models (LLM) to enhance chart datasets for training purposes.
4) The paper demonstrates good writing quality, encompassing recent literature developments and technical aspects. Notably, Table 1 is informative, and the related work section has covered relevant areas. Additionally, Figure 1 is thoughtfully organized and imparts valuable insights.
5) The experiments are well-considered, offering comprehensive results and insightful ablation studies.
6) The potential impact is substantial, given the practical utility of the proposed method, which addresses challenges not effectively addressed by ChatGPT or existing solutions.

**Weaknesses:**

1) The approach itself is a bit straightforward as it is not totally end-to-end. However, I don't think it is a drawback, especially given its novel setting compared to other literature.
2) Some typos: e.g. 4.3 ACHIEVING 100% PERFORMANCE BY ONLY 20% REAL DATA. The ending stop shall be removed.

**Questions:**

Can you compare your method with GPT-4V or other general multi-modality tools?

---

> ### Author Response · Authors · 2023-11-16
> **Response to Reviewer ojHH: Part 1/1**
>
> **Q1：The approach itself is a bit straightforward as it is not totally end-to-end. However, I don't think it is a drawback, especially given its novel setting compared to other literature.**
>
> **A1**: Thanks for your approval and insightful comment. We are honored that we have reached a consensus that our proposed two-stage pipeline is not a drawback. Furthermore, the main reason why two-stage pipeline is not a drawback is as follows:
> - Explicit chart image representations (Such CSV format or the proposed STR format) obtained using perception module can enhance the interpretability of the subsequent reasoning stage.
> - The first stage of the proposed perception-reasoing pipeline explicitly extracts the structured text to represent chart image (i.e., CIE task), which can be used as the pre-training corpus of the existing large language models or vision language models.
>
> &ensp;
>
> **Q2: Some typos: e.g. 4.3 ACHIEVING 100% PERFORMANCE BY ONLY 20% REAL DATA. The ending stop shall be removed.**
>
> **A2:**  We thank the reviewer very much for pointing this out, and we have corrected it in the revised manuscript.
>
> &ensp;
>
> **Q3：Can you compare your method with GPT-4V or other general multi-modality tools?**
>
> **A3:**  We would like to thank the reviewer for your interest in our work, and we also have a fascination with the performance of GPT-4V, since OpenAI has just recently opened up their API for GPT-4V.
>
> According to the reviewer's comments, we would like to show three visualization comparisons among StrcutChart, **GPT-4V** and **LLAVA-1.5** in the following link [https://drive.google.com/file/d/1BX7wqS792Z7N4uJNcBagSJbo5622wVYB/view?usp=drive_link]. Note that for QA task, our StructChart is restricted to generating answers only in order to report the quantitative results on ChartQA dataset.
>
> From the visualization results, it can be seen that the proposed StructChart has a strong Chart-based reasoning ability, showing promising performance on multiple downstream tasks.  Although GPT-4V also has a strong performance, our model weights, training code, simulation code, and simulation datasets will all be open-sourced for this community! Besides, we have supplemented the visualization comparison results on Page 24-26, Appendix I of the revised manuscript.

---

> ### Author Response · Authors · 2023-11-22
> **Paper Update**
>
> Dear Reviewer ojHH,
>
> Thanks a lot for your valuable comments, and constructive suggestions, and we have updated the revised paper.
>
> FYI, we provide a summary of the updates inspired by you:
>
> - We emphasize the motivations of designed two-stage pipeline in **Sec 3.1, Page 3** of the main text.
>
> - We fix the type error in **Sec 4.3, Page 7** in the main text, and check the similar typos.
>
> - We demonstrate visualization comparisons among StructChart, GPT-4V and LLAVA-1.5 in **Fig 13,14 and 15 (Page 24-26), Appendix I (Page 17).** We also hint at these comparisons in **Sec 4.4, Page 9** of the main text.
>
> We all think that your insightful suggestions make our paper more complete and convicing. Thanks again for your precious time.
>
> Best regards,
>
> Authors of Paper 1125

---

> ### Author Response · Authors · 2023-11-23
> **Looking forward to a reply**
>
> Dear Reviewer ojHH,
>
> We are very grateful to the reviewer for the insightful comment about the comparison with GPT-4V or other general vision-language methods.
>
> During the rebuttal period, we have supplemented the visualization comparisons among StrcutChart, GPT-4V and LLAVA-1.5 in the following link https://drive.google.com/file/d/1BX7wqS792Z7N4uJNcBagSJbo5622wVYB/view?usp=drive_link. Also, the comparison results have been included in Page 24-26, Appendix I of the updated manuscript.
>
> Thanks for your effort in improving the quality of our manuscript, and it is appreciated that you could consider these modifications and rapid actions of our manuscript.
>
> &ensp;
>
> Best regards,
>
> Authors of Paper 1125

---

### Official Review · Reviewer_mxZb · 2023-10-31

**Soundness:** 2 fair
**Presentation:** 3 good
**Contribution:** 2 fair
**Rating:** 5
**Confidence:** 3

**Summary:**

This paper introduces a StructChart methodology for extracting information from visual chart data to enhance downstream perception and reasoning tasks. To achieve this, the authors initially transform the chart data from Linearized Comma-Separated Values Tokens (LCT) into the proposed Structured Triplet Representations (STR), thereby establishing a connection between chart perception and reasoning. To qualitatively evaluate the chart perception tasks, the authors also introduce a Structuring Chart-oriented Representation Metric (SCRM). This metric evaluates the extracted chart data using Intersection over Union (IoU). Additionally, the authors construct a synthetic chart dataset called SimChart9K, which is helpful for downstream tasks. The numerical experiment shows the efficiency of the proposed method. However, I have some concerns about this paper. My detailed comments are as follows.

**Strengths:**

1. The authors seek to transform the chart information from LCT to STR, which bridges the gap between chart perception and reasoning.
2. The authors construct a synthetic dataset named SimChart9K by leveraging an LLM-based self-inspection data production scheme.
3. Experimental results on the ChartQA and PlotQA datasets demonstrate the effectiveness of the proposed StructChart method.

**Weaknesses:**

1. This paper emphasizes that STR reduces the task gap between chart perception and reasoning. However, the reasoning process is based on the black box GPT-3.5, which cannot find the relationship between STR and the reasoning process. More explanations are required.
2. As shown in Eqn. (2), the SRT splits each element in LCT as one unique sample. However, SRT may destroy the intrinsic relationship between the original elements, making them independent.
3. In Equation (3), the authors introduce the Entity Match method, which employs Intersection over Union (IOU) to compare predictions with ground-truth entities. Do the authors consider the order of the entity strings during the matching process? Ordinarily, aligning the strings in the correct order is essential for accurate matching. However, Equation (3) lacks a detailed explanation of how this matching process is carried out. More discussions are required.
4. The authors adopt STR to make each entity independent. How do the authors determine the correspondence between predictions and specific ground-truth?
5. In Table 2, it's noteworthy that the authors have not included a comparison with Matcha[1] and Deplot[2] on the Chart2Text[3] dataset. An explanation for this omission is needed.
6. In Table 2, the comparisons between StructChart and the compared methods (Matcha and Deplot) somewhat is unfair. StructChart leverages powerful GPT-3.5 as the reasoning model, whereas Matcha is based on Pix2Struct[4] and Deplot relies on Codex or GPT-3. Thus, the disparities in performance between StructChart and the compared methods could potentially stem from the utilization of different reasoning models (i.e., GPT-3 and GPT-3.5).
7. In Table 4, the authors conduct a comparison of StructChart with various baseline methods using the Exact Match metric. It's important to note, however, that the authors have omitted a direct comparison with Deplot, which outperforms StructChart with a higher score (i.e., 76.7 compared to 65.3). In this way, StructChart has no advantage compared with Deplot even if it uses more powerful GPT-3.5.
8. In Table 2, the ‘merging’ is confusing. Does it merge the ChartQA with the SimChart 9K?


[1] Matcha: Enhancing visual language pretraining with math reasoning and chart derendering, ACL 2023.

[2] Deplot: One-shot visual language reasoning by plot-to-table translation, ACL Findings 2023.

[3] Chart-to-text: Generating natural language descriptions for charts by adapting the transformer model, ACL 2020.

[4] Pix2Struct: Screenshot Parsing as Pretraining for Visual Language Understanding. ICML 2023.

**Questions:**

Please refer to the Weaknesses

---

> ### Author Response · Authors · 2023-11-15
> **Response to Reviewer mxZb: Part 1/3**
>
> **Q1: The reasoning process is based on the black box GPT-3.5, which cannot find the relationship between STR and the reasoning process**
>
> **A1:**
> We are very grateful to the reviewer for the insightful comments. We agree with the reviewer that the reasoning module is based on the black box, but the proposed Structured Triplet Representations (STR) can be regarded as the input of such a reasoning module. As stated in Section 3.1 of the main text, our StructChart employs the two-stage method including perception and reasoning stages, respectively. The perception module generates the STR, and then, the generated STR can be regarded as the input of the reasoning module (GPT-3.5).  Besides, due to that the generated STR can effectively represent the positional relations between row and column headers of a given chart, the perception-reasoning task gap can be effectively alleviated.
>
> Experimentally, we have verified that the proposed STR representations are beneficial for improving reasoning ability on different downstream tasks. For example, in QA task, the results in the following Table show that, STR outperforms LCT by 9.0% evaluated on ChartQA validation set (all the settings in GPT-3.5 using LCT input and STR input are consistent, including prompt, code version and parameters). For other downstream tasks including summarization and redrawing tasks, we visually illustrate the comparisons between LCT and STR  in Appendix E (please see Figs 11 and 12), where a comma is introduced as noise as illustrated in Fig. 11, and the separator comma itself is included as illustrated in Fig. 12. Both of the case indicate that, STR is more friendly for reasoning process than LCT.
>
> | Input Representation | Method | Train on | QA val |
> |:---:|:---:|:---:|:---:|
> | LCT | StructChart+GPT3.5 | ChartQA+SimChart9K | 56.3 |
> | **STR** | StructChart+GPT3.5 | ChartQA+SimChart9K | **65.3** |
>
> &ensp;
>
> **Q2: STR may destroy the intrinsic relationship between the original elements, making them independent**
>
> **A2:** Thanks. First of all, the proposed Structured Triplet Representations (STR)  does not destroy the intrinsic relationship between the original elements, due to the following reasons:
>
> In Section 3 of the main text, we claim that, "The extracted intermediate CSV text is structured into a triplet form to elucidate the intricate position relationship between the header and index" **In this case**, "header" indicates column header in CSV format while "index" refers to row header. For STR transformation, each related column header $Entity_{c_m}$ and row header $Entity_{r_n}$ will form a tuple with their relationship $Value_{r_n}^{c_m}$, and the set composed of all tuples aims to represent a complete chart image. Hence, the intrinsic relationship between the original entities will not be destroyed.
>
> &ensp;
>
> **Q3: Do the authors consider the order of the entity strings during the matching process? Eq (3) lacks a detailed explanation of how this matching process is carried out**
>
> **A3:**
> Thanks for your valuable comments. In Eq. 3 of the main text, the order of two Entity strings $Entity_{r_n}$ and $Entity_{c_m}$ in each Tuple $(Entity_{r_n}, Entity_{c_m}, Value_{r_n}^{c_m})$ is important for matching. For example, the two Tuples $(Alice, Bob, 100)$ and $(Bob, Alice, 100)$ should be matched exactly. **To ensure the order of Entity string does not affect the matching process**, we reorder both two Entity strings according to the following pre-process:
> * Lowercase all characters in Entity strings.
> * The two Entity strings in each Tuple are sorted by the ASCII of the first letter.
> * If the ASCII of the first letter is the same, the second letter is compared, etc.
>
> In the revised manuscripts, we have supplemented the detailed explanation of how to perform the matching process in Appendix B on Page 14.
>
> &ensp;
>
> **Q4: How do the authors determine the correspondence between predictions and specific ground-truth?**
>
>
> **A4:** Thanks. For all datasets mentioned in the main text, we have converted Ground-Truth (GT) into the proposed STR format during the matching process. Hence, the matching process is conducted between the STR prediction and the STR GT to determine the correspondence between predictions and specific ground-truth. For example, given that both prediction and GT are sets in the form of Tuples $(Entity_{r_n}, Entity_{c_m}, Value_{r_n}^{c_m})$, the matching process is as follows:
>  - Query whether each Tuple in GT set has a matching Tuple in prediction set.
>  - When judging whether two Tuples match, calculate the similarity for Entity string and Value string according to Eqs 3 and 4 in the main text, respectively (the confusion about the order of Entity string has been claimed in the response to Q3).
>  - Calculate the Intersection over Union (IoU) between prediction set and GT set according to Eq. 7 in the main text. If the calculated IoU is larger than the preset threshold, the prediction and GT are matched.

---

> ### Author Response · Authors · 2023-11-15
> **Response to Reviewer mxZb: Part 2/3**
>
> **Q5: In Table 2, why the authors have not included a comparison with Matcha and Deplot on the Chart2Text dataset.**
>
> **A5:**
> We thank the reviewer very much for pointing this out. The chart images in Chart2Text dataset contain complex background information from the website. As a result, in Table 2,  employing the Matcha and Deplot can only achieve very low perception performance (almost close to 0) for Chart-oriented Information Extractor (CIE) task, due to the complex chart background.
>
> For example, on perception task, Matcha can only output the content in the form of HTML format, while Deplot often outputs noisy backgrounds (e.g. website page environment containing other irrelevant text).  In contrast, the proposed Structchart can extract accurate information of the chart from complicated background, such as information from the website. Specifically, we have illustrated visualization comparisons among Matcha, Deplot, and our StructChart for CIE task on Chart2Text dataset in **Figure 6,7,8 of Appendix F**.
>
>
> &ensp;
>
>
> **Q6: In Table 2, the comparisons between StructChart and the compared methods (Matcha and Deplot) somewhat is unfair, since StructChart leverages powerful GPT-3.5 as the reasoning model, whereas Matcha is based on Pix2Struct and Deplot relies on Codex or GPT-3.**
>
>
> **A6:** We thank the reviewer very much for pointing this out. First, we would like to emphasize that In Table 2, we only report the results of chart perception task using our proposed SCRM metric. For perception part, all comparisons with Matcha and Deplot are fair, since we directly evaluate the well-trained Matcha and Deplot on the ChartQA, PlotQA, and Chart2Text, where all evaluation settings are **consistent** with the proposed StructChart.
>
> Furthermore, for reasoning part, Matcha is an end-to-end model (without the help of GPT-3 or 3.5), and we cannot modify its reasoning module (including network design and language model selection) during the evaluation process. As a result, we only choose to directly compare with Matcha. Experimental results show that our StructChart has a 1.1% lead over Matcha in average accuracy. Besides, according to the Reviewer's comment, to make a fair comparison, we supplement experiments of employing Deplot+GPT-3.5 to perform the QA task on ChartQA dataset, where we all use GPT-3.5 and ensure that the model settings (prompts, temperature, top-k, etc.) are consistent. The results in the following Table show that, our StructChart+GPT-3.5 outperforms Deplot+GPT-3.5 by 12.4% for QA task on ChartQA. Besides, It should be noted that there is no extra prompt engineering strategy (Chain-of-Thought, Program-of-Thought) nor evaluation tactics (self-consistency) during the reasoning process.
>
>
> |  Input Representations |         Model               | Train Set           | aug.  | human | avg.  |
> |:-----:|--------------|:---------------------:|:------:|:-------:|:------:|
> |  \   | Matcha              | ChartQA+PlotQA+C.D. | 90.2 | 38.2  | 64.2 |
> |  \  | Deplot+GPT-3.5      | ChartQA+PlotQA+C.D. | 69.3 | 36.6  | 52.9 |
> | LCT | StructChart+GPT-3.5 | ChartQA+SimChart9k  | 71.3 | 41.2  | 56.3 |
> | STR | StructChart+GPT-3.5 | ChartQA+SimChart9k  | **83.9** | **46.7**  | **65.3** |
>
>
> In the revised manuscripts, we have supplemented the corresponding experimental results and empirical analyses of Deplot+GPT3.5 in Table 6 of Appendix C on Page 15.

---

> ### Author Response · Authors · 2023-11-15
> **Response to Reviewer mxZb: Part 3/3**
>
> **Q7: In Table 4, the authors have omitted a direct comparison with Deplot, which outperforms StructChart with a higher score (i.e., 76.7 compared to 65.3).**
>
> **A7:** Thanks for this valuable comment. We would like to emphasize: 1) Deplot uses closed-source dataset for training. In contrast, our training set including ChartQA and SimChart9K is open-source; 2) Our simulated data, SimChart9k, is scalable and we have verified through experiments that the more simulation data, the better the performance of perception stage in Table 3 of the main text.
>
> Besides, the reason why StructChart has no advantage compared with Deplot (i.e., 76.7 compared to 65.3) can be concluded:
> - During inference, Deplot applies Chain-of-Thought (CoT) or Program-of-Thought (PoT) framework as prompt engineering to help the reasoning module boost the performance, but we did not employ any prompt engineering techniques. Since we cannot access specific prompt designs in Deplot, the fairness of result comparison on QA task cannot be guaranteed.
> - For evaluation, Deplot adopts Self-Consistency (SC) strategy (generate multiple reasoning paths and answers, and finally select the one with the most frequent answers as the final answer output). However, to ensure consistent output results given the same input condition, we set the temperature parameter of GPT-3.5 as 0.
>
> Finally, for fair comparison, we evaluate StructChart and Deplot with the same reasoning model (GPT-3.5) setting without any prompt engineering strategies nor evaluation tactics. As reported in the following Table, our StructChart has a 12.4% lead over Deplot for QA task on ChartQA.
>
>
> |  Input Representations   | Model               | Train Set           | aug.  | human | avg.  |
> |:-----:|--------------------|:---------------------:|:------:|:-------:|:------:|
> |   \  | Deplot+GPT-3.5      | ChartQA+PlotQA+C.D. | 69.3 | 36.6  | 52.9 |
> | STR | StructChart+GPT-3.5 | ChartQA+SimChart9k  | **83.9** | **46.7**  | **65.3** |
>
> &ensp;
>
> **Q8: Confusion about 'merging' in Table 2**
>
> **A8:** Thanks for this valuable comment. Merging-set in Table 2 refers to that, training samples are merged from the real datasets, including ChartQ, PlotQA, and Chart2Text.
> The purpose of such a merging-set setting is : 1) to better understand the performance scalability given more training data from real-world domain rather than simulation domain, and 2) to provide a baseline that is trained from real-world domain, and such baseline can be used for comparing with that trained from our proposed SimChart9K for chart perception task.
>
>
> | Val Set | Model       | Train Set          | mprecision (strict) |  mprecision (slight)      |   mprecision (high)     |
> |:-------:|:-----------:|-----------------|:-------:|:--------:|:--------:|
> | ChartQA | StructChart | Merging            | 0.7017     | 0.8227 | 0.8591 |
> | ChartQA | StructChart | ChartQA+SimChart9K | 0.7116    | 0.8182 | 0.8527 |
>
>
> Results from Table 2 in the main text demonstrate that, StructChart continuously improves the perception performance of chart data on each domain given more training samples.
>
> According to the reivewer's comment, we have supplemented some descriptions of the merging setting in the caption of Table 2 in the revised manuscript, to avoid confusion.

---

> ### Author Response · Authors · 2023-11-20
> **Look forward to further discussion.**
>
> Dear Reviewer mxZb,
>
> We have made a rapid response to your concerns, and hope that our response can address your concerns. As the deadline for discussion is approaching, we would greatly appreciate it if you could let us know if there are any other questions or issues regarding the paper or our response. We are looking forward to further discussion.
>
> Best regards,
>
> Authors of Submission 1125

---

> ### Author Response · Authors · 2023-11-22
> **Paper Update**
>
> Dear Reviewer mxZb,
>
> Thanks a lot for your valuable comments, and constructive suggestions, and we have updated the revised paper.
>
> FYI, we provide a summary of the updates inspired by you:
>
> - For confusion may caused by matching process, we explain in detail in **Appendix B, Page 13**. Also, we make the hint in **Sec 3.2, Page 4** of the main text.
>
> - For unfair comparisons between StructChart with other methods, we claim the settings we adopted in GPT-3.5 in **Sec 4.1, Page 6** of the main text, as well as further discussion in **Appendix C, Page 14**. The QA result for Deplot with fair settings is supplemented in **Table 4, Page 8** of the main text.
>
> - We redefine "Real maerging" and "Real & Sim Merging" with clear explanation in **Table 2, Page 7** of the main text.
>
> We all think that your insightful suggestions make our paper more complete and convicing. Thanks again for your precious time.
>
> Best regards,
>
> Authors of Paper 1125

---

> > ### Author Response · Authors · 2023-11-23
> > **Looking forward to a reply**
> >
> > Dear Reviewer mxZb,
> >
> > &ensp;
> >
> > We greatly appreciate the time you've invested in reviewing our response. Having submitted our rebuttal, we are eager to know if our response has addressed your concern. As the end of the rebuttal phase is approaching, we look forward to hearing from you for any further clarification that you might require, in order to further revise our paper and improve the quality of the manuscript.
> >
> > &ensp;
> >
> > Best regards,
> >
> > Submission 1125 authors

---

### Author Response · Authors · 2023-11-22
**General Response**

Dear Area Chairs and Reviewers,

We greatly appreciate the reviewers’ time, valuable comments, and constructive suggestions. Overall, the Reviewers (mxZb, ojHH, Rkaq) appreciate the significant experimental effectiveness of StructChart both in chart perception task and various downstream tasks, as well as the substantial impact it brings to the field. They highlight the proposed STR representation and SCRM is novel and applicable for chart-related tasks (mxZb, ojHH). The experiments are considered informative, well-considered, comprehensive and insightful (ojHH). They also appreciate the excellent contribution (Rkaq) and presentation (mxZb, ojHH, Rkaq).

In the author response period, we make every effort to address reviewers’ concerns and provide additional experimental results to further verify our contributions. Here is a summary of what we have done in the rebuttal phase:
- We provide a detailed explanation of the matching process in Structuring Chart-oriented Representation Metric (SCRM).
- We emphasize the settings adopted in the reasoning modele (GPT-3.5) for different methods in QA task, as well as the reason why the settings were adopted.
- We supplement the generalizability study of the conclusions/findings claimed in the main text with experiments on more benchmarks.
- We summarize the motivations for designing a perception-reasoning pipeline, rather than an end-to-end one.
- We supplement the visualization comparisons among the performance of StrcutChart, GPT-4V and LLAVA-1.5 on chart-related downstream tasks.
- We improve some "confusing" expressions and type errors in revision.

Thank you again for your precious time on the review. We hope that our response has addressed your concerns. We are happy to have further discussion on anything unclear about our paper.


Best regards,

Authors of paper 1125

---

### Author Response · Authors · 2023-11-22
**Paper Update**

Dear AC and Reviewers,

Thank you for your precious time and constructive suggestions to improve the quality of our manuscript. We uploaded a new version of our paper and the updates made during the rebuttal period are highlighted in blue in the revision. We provide a summary below:

**[mxZb]**  For confusion may caused by **matching process**, we explain in detail in **Appendix B, Page 13**. Also, we make the hint in **Sec 3.2, Page 4** of the main text.

**[mxZb]**  For **unfair comparisons** between StructChart with other methods, we claim the settings we adopted in GPT-3.5 in **Sec 4.1, Page 6** of the main text, as well as further discussion in **Appendix C, Page 14**. The QA result for Deplot with fair settings is supplemented in **Table 4, Page 8** of the main text.

**[mxZb]** We redefine "Real maerging" and "Real & Sim Merging" with clear explanation in **Table 2, Page 7** of the main text.

**[ojHH]** We emphasize the **motivations** of designed **two-stage** pipeline in **Sec 3.1, Page 3** of the main text.

**[ojHH]** We fix the **type error** in **Sec 4.3, Page 7** in the main text, and check the similar typos.

**[ojHH]** We demonstrate **visualization comparisons** among StructChart, GPT-4V and LLAVA-1.5 in **Fig 13,14 and 15 (Page 24-26), Appendix I (Page 17)**. We also hint at these comparisons in **Sec 4.4, Page 9** of the main text.

**[Rkaq]** We emphasize the **generalizability** of the findings (a larger scale of the data will lead to better performance for CIE task) in **Sec 4.2, Page 6** of the main text. Also, we supplement the corresponding generalizability experiments in **Table 2, Page 7** of the main text.

**[Rkaq]** We emphasize the **generalizability** of the findings (“we can achieve a high-performance CIE only leveraging few-shot real samples”) in **Sec 4.3, Page 8** of the main text. Also, we supplement the corresponding generalizability experiments in **Table 6, Page 9** of the main text.

**[Rkaq]** For benchmark **FigireQA**, we provide a brief description in **Sec 4.1, Page 6** of the main text. Meanwhile, we compare FigureQA among other datasets we adopted in this work in **Appendix D, Page 14**. Also, the QA results in FigureQA are shown in **Table 7, Page 14** of the Appendix.

We hope that our revision makes our presentation clearer and covers reviewers' concerns. We are happy to discuss any remaining questions about our work.

Best regards,

Authors of Paper 1125

---

### Meta-Review · Area_Chair_F7n4 · 2023-12-05

**Metareview:**

While the reviewers appreciate the goal and dataset, there are many questions about the method and experiments, e.g. about lack of interpretability, missing comparisons, and limited method innovation, that caused lukewarm reviewer scores.

**Justification For Why Not Higher Score:**

No scores above 6

**Justification For Why Not Lower Score:**

n/a

---

### Decision · Program_Chairs · 2024-01-16

Reject